# Genome-scale chromatin binding dynamics of RNA Polymerase II general transcription machinery components

Kristyna Kupkova [1,2], Savera J Shetty[1], Elizabeth A Hoffman[1], Stefan Bekiranov[1] & David T Auble [1✉]

## Abstract

A great deal of work has revealed, in structural detail, the components of the preinitiation complex (PIC) machinery required for initiation of mRNA gene transcription by RNA polymerase II (Pol II). However, less-well understood are the in vivo PIC assembly pathways and their kinetics, an understanding of which is vital for determining how rates of in vivo RNA synthesis are established. We used competition ChIP in budding yeast to obtain genome-scale estimates of the residence times for five general transcription factors (GTFs): TBP, TFIIA, TFIIB, TFIIE and TFIIF. While many GTF-chromatin interactions were short-lived (< 1 min), there were numerous interactions with residence times in the range of several minutes. Sets of genes with a shared function also shared similar patterns of GTF kinetic behavior. TFIIE, a GTF that enters the PIC late in the assembly process, had residence times correlated with RNA synthesis rates. The datasets and results reported here provide kinetic information for most of the Pol II-driven genes in this organism, offering a rich resource for exploring the mechanistic relationships between PIC assembly, gene regulation, and transcription.

**Keywords** Competition ChIP; Kinetics; Preinitiation Complex; Synthesis Rates
**Subject Categories** Chromatin, Transcription & Genomics; RNA Biology

## Introduction

Transcription is a highly complex biochemical process whose exquisite regulation is of fundamental importance in determining cell function and fate. A tremendous amount of information is available on the structure, biochemical functions, and relationships of various transcription factors (TFs), cofactors, and subunits of the general transcription machinery (He et al, 2016; Plaschka et al, 2016; Sainsbury et al, 2015; Hahn and Young, 2011; Hahn, 2004). This includes structures of the transcription preinitiation complex (PIC), which assembles at promoters and consists of the general transcription factors (GTFs) TFIIA, TFIIB, TFIID, TFIIE, TFIIF, and TFIIH, as well as RNA polymerase II (Pol II) (Patel et al, 2018; Plaschka et al, 2016; He et al, 2016; Ehara et al, 2017; Sainsbury et al, 2015; Hahn and Buratowski, 2016; Nogales et al, 2017). As these factors participate in all Pol II-mediated transcription in yeast, we use the term "PIC" in this paper to refer to complexes that catalyze the initiation of transcription, including during the first round of synthesis upon promoter activation as well as reinitiation or ongoing initiation at unregulated promoters. In addition, genome-wide analyses have provided global snapshots of many factors along the eukaryotic DNA template (Venters et al, 2011; Harbison et al, 2004; Yen et al, 2012; Rossi et al, 2021). These combined studies have led to a conceptual framework in which PICs are assembled stepwise at promoters. This process begins with nucleation by TFIID, a multisubunit complex that contains the DNA-binding subunit TATA-binding protein (TBP) (Patel et al, 2020; Louder et al, 2016), and can be further facilitated by the binding of TFs and coactivators that physically contact GTFs (Chen & Pugh, 2021). In vitro, following the binding of TBP/TFIID to a TATA-containing promoter, TFIIA and TFIIB can then associate with the complex, followed by Pol II in association with TFIIF, and then TFIIE (Farnung and Vos, 2022). This multisubunit complex provides the substrate for the recruitment of TFIIH (Tsutakawa et al, 2020), whose activities are required in vivo but may be dispensable in vitro using naked DNA substrates (Rimel and Taatjes, 2018). A key factor contributing to PIC assembly in vivo is the Mediator, which physically contacts multiple GTFs and modulates the activities of TFIIH (Rimel and Taatjes, 2018; Schilbach et al, 2017; Malik et al, 2017). Live-cell imaging has documented the dynamic behavior of these factors and is generally consistent with such an assembly pathway, albeit occurring via highly dynamic and short-lived complexes (Nguyen et al, 2021). Importantly, the understanding of PIC assembly has emerged mainly from studies that have focused on the analysis of stable complexes formed in vitro or identified in vivo, lacking information about the locus-specific dynamics of the process. Furthermore, some evidence suggests that the canonical in vitro assembly pathway may not apply to PICs at all promoters in vivo (Guglielmi et al, 2013; Luse, 2014; Sikorski and Buratowski, 2009; Baek et al, 2021; Nguyen et al, 2021). In addition to unexplored assembly pathway complexity, it has become apparent that in vivo transcription is a highly dynamic and stochastic process, with RNA synthesis often occurring from individual genes in bursts, and

[1]Department of Biochemistry and Molecular Genetics, University of Virginia Health System, Charlottesville, VA 22908, USA. [2]Center for Public Health Genomics, University of Virginia Health System, Charlottesville, VA 22908, USA. ✉E-mail: auble@virginia.edu

**Glossary**

| | | | | |
|---|---|---|---|---|
| CC | Competition ChIP | | PC2 | second principal component |
| DBF | DNA binding factor | | PCA | principal component analysis |
| DTA | dynamic transcriptome analysis | | PIC | preinitiation complex |
| FDR | false discovery rate | | Pol II | RNA polymerase II |
| GTF | general transcription factor | | TBP | TATA-binding protein |
| GO:BP | gene ontology biological process | | TE | transcription efficiency |
| GO:MF | gene ontology molecular function | | TF | transcription factor |
| GTF | general transcription factor | | TSS | transcription start site |
| padj | adjusted p-value | | WP | WikiPathways |
| PC1 | first principal component | | | |

with variability occurring among genetically identical cells (Sanchez and Golding, 2013; Raser and O'Shea, 2004). Most models of RNA expression based on these types of observations do not posit particular features of protein–DNA complex behavior as the explanation, and relatively few genes have been analyzed in depth (Boeger et al, 2008; Brown and Boeger, 2014; Lenstra et al, 2016; Boettiger et al, 2011; Ravarani et al, 2016). Indeed, live cell imaging approaches have revealed that while TFs, in general, display very dynamic interactions with chromatin, the functional consequences of their interaction kinetics are only beginning to be explored on a mechanistic level (Hager et al, 2009; Voss and Hager, 2014; Nguyen et al, 2021).

The premise of this study is that PIC assembly dynamics are variable across the genome and that the identification of kinetic pathways in PIC assembly will shed light on mechanisms of regulation that operate at the level of transcription initiation. To better understand PIC assembly in vivo, we have used an approach called competition-chromatin immunoprecipitation (competition ChIP, ref. (Lickwar et al, 2013)) to measure the site-specific, genome-scale chromatin binding dynamics of five GTFs (TBP, TFIIA, TFIIB, TFIIE, and TFIIF) in the budding yeast *S. cerevisiae*. In addition, we compared promoter binding dynamics of these factors with RNA synthesis rates to determine how chromatin binding of key PIC components relates to the production of RNA. To our knowledge, this represents the first comprehensive analysis of PIC dynamics, provides a global picture of PIC assembly, and highlights promoter-specific variation.

## Results

Competition ChIP (CC) is an approach in which cells harbor two isoforms of a transcription factor of interest with distinguishable epitope tags (Fig. 1A). We engineered diploid yeast cells to constitutively express one isoform with a Myc tag under control of the endogenous promoter and with the second isoform tagged with HA and under inducible *GAL* promoter control. In the CC experiments, cells were shifted to galactose at time zero to induce expression of the HA-tagged competitor isoform, followed by cell culture sample collection at various time points (Fig. 1B). We then measured the relative occupancies of the Myc- and HA-tagged species genome-wide at each time point (Fig. 1C) and used the relative occupancies as input to a model that describes the competition for chromatin binding to each site, yielding the site-specific residence time (Fig. 1D). The principle of the assay is outlined in Fig. 1E,F, which illustrate how the occupancy ratios of the two isoforms of a particular factor would change if the factor

has a short or long residence time at a particular site. Notably, TFIIA, TFIIE, and TFIIF are biochemically composed of more than one subunit, and thus, for these factors, we epitope-tagged one subunit and placed one copy of each subunit under *GAL* control in order to induce balanced expression when cells were grown in galactose (see Materials and Methods).

For each factor, we first measured the levels of both isoforms by Western blotting (Figs. 2A–C and EV1; Appendix Table S1). The time-dependent accumulation of competitor isoforms displayed cooperative induction consistent with a Hill equation (Estrada et al, 2016) with induction half-times of ~43 min and Hill coefficients of ~4.5 on average (Fig. 2B,C). We estimated residence times by fitting the normalized time-dependent turnover ratios to a turnover model (Zaidi et al, 2017a) (see Materials and Methods), and compared the fits to the HA-tagged competitor's synthesis rate. In this way, we were able to assign residence times for binding interactions with significantly longer (>1 min) rates of turnover compared to the rate of competitor synthesis, and for reliable fits that were not significantly different from the rate of competitor induction, we were able to classify the chromatin binding residence times as <1 min (see Materials and Methods). Overall, we were able to estimate residence times for each GTF binding to ~3000 or more promoters (Fig. 2D; Dataset EV1). This represents roughly half of the Pol II promoters in the *S. cerevisiae* genome. Representative fits are shown in Fig. 2E; Appendix Fig. S1. Note that the HA/Myc ratios at sites with rapid turnover closely mimic the time course of competitor induction, whereas more long-lived complexes have turnover ratios that are notably displaced to the right of the competitor induction curves. The distributions of turnover times are shown in Fig. 2F. We identified different numbers of sites for each TF for which we were able to assign residence times; this is indicative of differences in the number of sites for which we were able to obtain reliable fits of the kinetic data, as well as likely differences in the efficiency of formaldehyde capture of short-lived complexes. It is notable that the majority of TBP, TFIIA, TFIIB, and TFIIF chromatin interactions were short-lived (i.e., <1–2 min), whereas the majority of TFIIE complexes displayed residence times in the several-minute range. It was also notable that TFIIF residence times were bimodal, with most estimates being short-lived (~2 min or less) and the rest in the 5–10 min range (Fig. 2F; discussed below).

To determine the relationship between GTF promoter residence time and the rate of RNA synthesis from the corresponding genes, we measured newly synthesized RNA under these same conditions (Fig. EV2A; Dataset EV2). Replicate samples (n = 2) were acquired at 20 and 60 min post galactose-induction. There was excellent agreement between the replicates and between the two time points

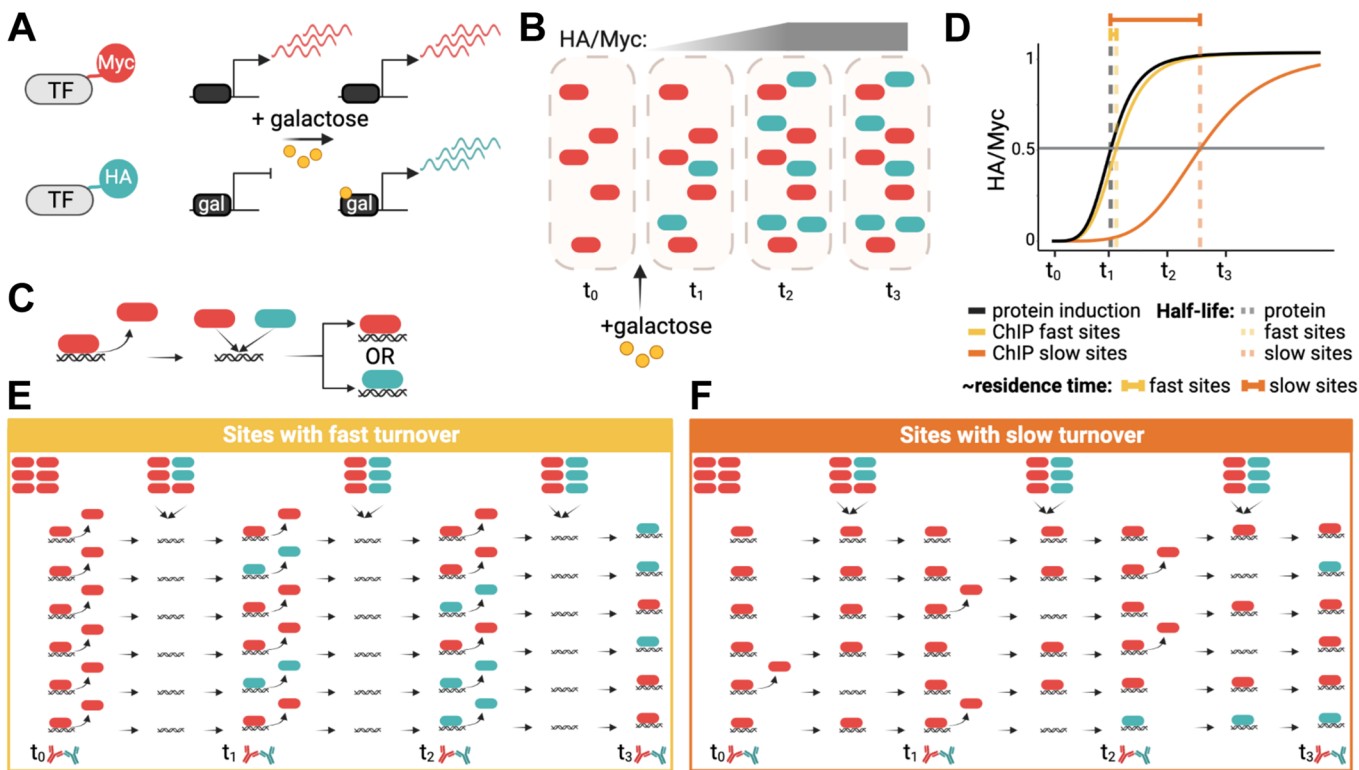

**Figure 1.  Competition ChIP overview.**

(A) A Myc-tagged isoform of a TF is expressed constitutively using the endogenous promoter, while an HA-isoform is expressed under the control of a galactose-inducible promoter. (B) Illustration showing protein induction upon adding galactose. The HA/Myc ratio increases over time until it reaches saturation. (C) Once a given TF unbinds DNA, the two isoforms compete for binding to the available site. (D–F) D shows a simplified illustration of residence time estimation based on the lag of the normalized HA/Myc ChIP signal ratio relative to the competitor protein induction curve, as further illustrated in (E) for sites with fast turnover and (F) for sites with slow turnover. In (E,F), the icons in the top row indicate relative levels of constitutive (red) and competitor (green) isoforms.

(Figs. 3A and EV2B–D). Dynamic transcriptome analysis (DTA, ref. (Miller et al, 2011)) was applied to estimate RNA synthesis rates (Fig. EV2E), which were in reasonable concordance with earlier data from cells grown in galactose (Fig. EV2F, ref. (García-Martínez et al, 2004)). We divided the mRNA synthesis rates into quartiles and compared them to GTF residence times (Fig. 3B,C). Residence times for TFIIA and TFIIB were, on average, modestly shorter for highly expressed genes compared to genes with lower expression levels, which may suggest a kinetic bottleneck in PIC assembly for poorly expressed genes that occurs after the binding of these two factors (see Discussion). Strikingly, the average TFIIE residence time increased with gene expression level across these four groups of genes (Fig. 3C), suggesting that the TFIIE residence time is an indicator of gene expression level. To relate residence time to RNA synthesis more directly, we calculated the ratio of mRNA molecules made per GTF binding event, which we previously defined as transcription efficiency (TE, ref. (Zaidi et al, 2017a)). TE was, on average, <1 mRNA synthesized per binding event for TBP, TFIIA, TFIIB, and TFIIF (Fig. 3D), suggesting that binding events by these factors do not efficiently give rise to the synthesis of mRNA. In addition, the TE values increased gradually and progressively for these factors, with TBP having the lowest TEs and TFIIE the highest, in line with the in vitro assembly pathway in which TBP binds to promoters first, followed by TFIIA and TFIIB, which

provide a platform for binding of TFIIF in association with Pol II (Farnung and Vos, 2022). Notably, the median TE for TFIIE was close to one, suggesting that the binding of TFIIE to promoters was associated with the production of one mRNA molecule on average. The results suggest that PIC formation is an increasingly efficient process along a pathway from TBP to TFIIE, and that the assembly of a TFIIE-containing PIC is associated with the production of a single molecule of mRNA. Using all of the GTF residence time data for Principal Component Analysis (PCA) revealed a correlation between GTF binding dynamics and RNA synthesis along the first principal component, PC1 (Fig. 3E; Appendix Fig. S2, where sites with <1 min residence times, which were randomly generated between 0–1 min, were excluded). This correlation can be appreciated quantitatively via the proportion of variance explained and visually by the distribution of color across the plot. To investigate the nature of this relationship in more detail, Pearson's correlation coefficients were computed between each GTF and PC1/PC2 (Fig. 3F) and between transcription rates and PC1/PC2 (Fig. 3G). The results show that the overall pattern was driven mainly by the positive correlations between TFIIE/TFIIF and RNA synthesis rate (Fig. 3F,G). This conclusion was further supported by linear modeling of the GTF residence time contributions to transcription rates (Fig. EV3A,B).

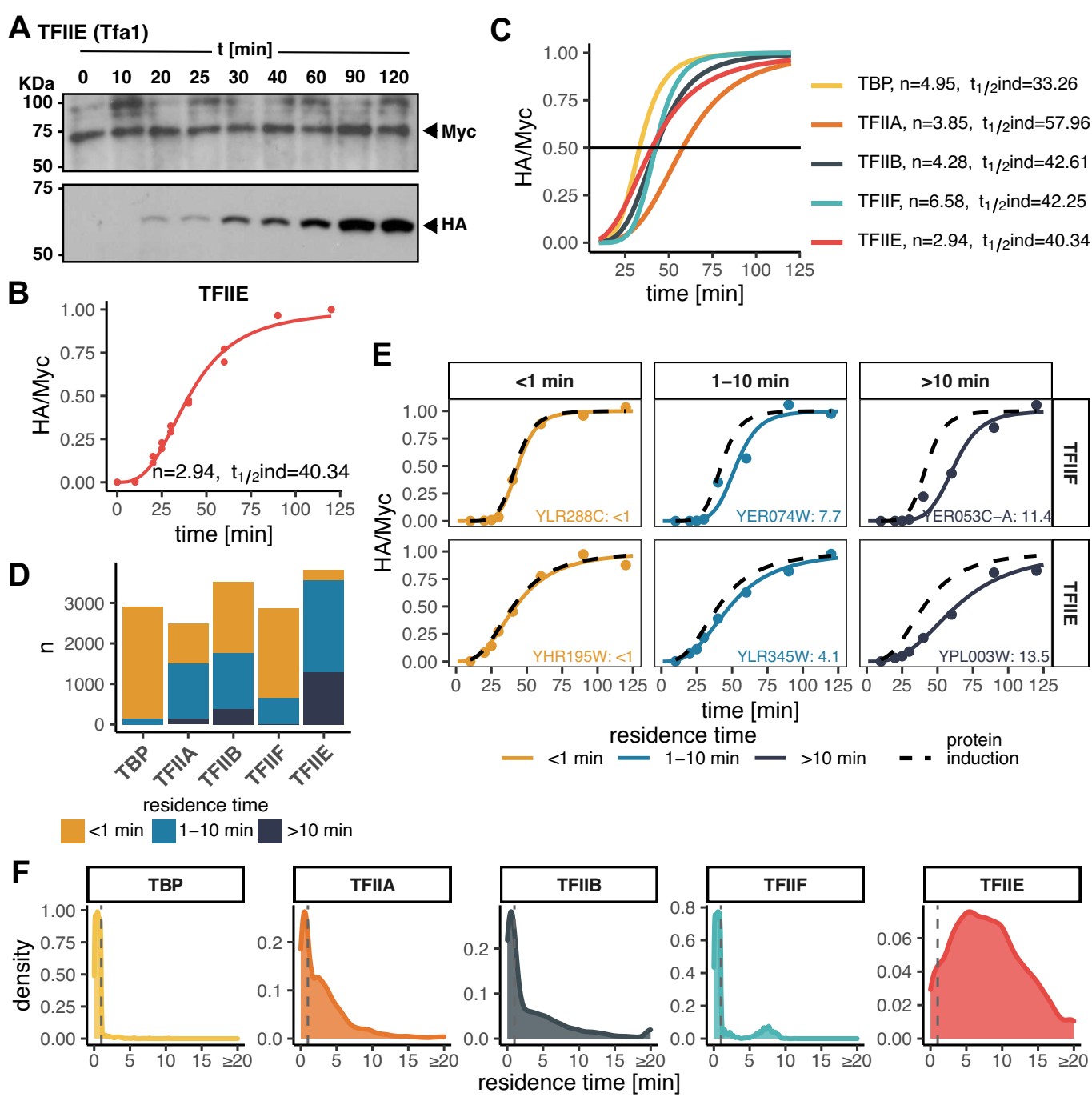

**Figure 2. GTF residence times.**

(A) TFIIE Western blots showing the isoform levels of the TFIIE subunit over the indicated time course. Galactose was added at $t = 0$ min. (B) Quantified Western blots from (A). Shown are normalized HA/Myc ratios ($n = 2$ biological replicates). The induction curve was fitted with a Hill coefficient (n) and induction half-time ($t_{1/2}$ind) as indicated. (C) Induction curves as in (B) for all targeted TFs with fit parameters indicated on the right. TFIIA (Toa1 subunit), TFIIE (Tfa1 subunit), and TFIIF (Tfg2 subunit). (D) Bar plot showing the number of sites (y-axis) categorized based on estimated residence time for each TF (x-axis). (E) Examples of sites with fast (<1 min), moderate (1–10 min) and slow (>10 min) turnover for TFIIF and TFIIE. Black dashed curves represent protein induction curves from (C), in color are shown the normalized HA/Myc ChIP signals (mapped reads) along with the fitted model. Gene target names, along with the estimated residence times in minutes, are included. (F) Distribution of estimated residence times for all GTFs. Values for reliably fast sites (<1 min) were randomly generated for plotting purposes and are separated by dashed lines. Density on the y-axis denotes the kernel density estimates used to approximate the frequency of a given residence time. Source data are available online for this figure.

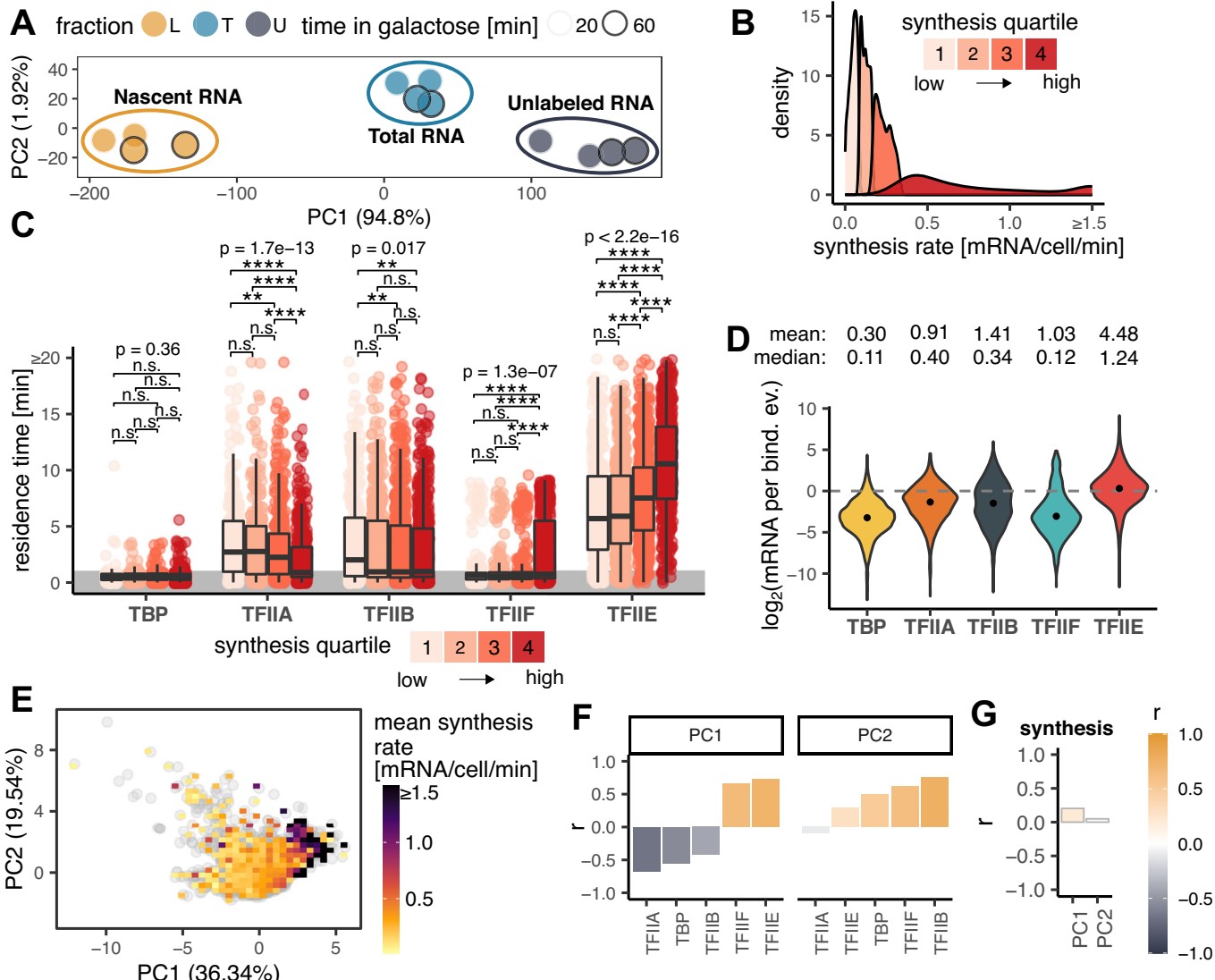

**Figure 3. Relationship between residence GTF residence times and synthesis rates.**

(A) PCA plot showing a low-dimensional representation of dynamic transcriptome analysis (DTA) samples without negative control. (B) Distribution of mRNA synthesis rate values measured by DTA separated into synthesis quartiles. (C) Box plots showing residence time distributions (y-axis) for all GTFs (x-axis) within the indicated synthesis quartile. Values for reliably fast sites (<1 min) were randomly generated for plotting purposes and are highlighted by the gray area. The middle line represents the median, the lower and upper edges of the boxes represent the first and third quartiles, and the whiskers represent the 1.5 * interquartile range. *P* values represent the results of Kruskal–Wallis tests for a given GTF. *P* value symbols (Wilcoxon test): n.s. $p \geq 0.05$, $*p \leq 0.05$, $**p \leq 0.01$, $***p < 0.001$, $****p \leq 0.0001$. Number of observations (*n*): TBP- Q1: 216, Q2: 324, Q3: 427, Q4: 605; TFIIA- Q1: 239, Q2: 320, Q3: 397, Q4: 426; TFIIB Q1: 302, Q2: 486, Q3: 581, Q4: 606; TFIIF- Q1: 253, Q2: 358, Q3: 432, Q4: 517; TFIIE- Q1: 416, Q2: 584, Q3: 610, Q4: 553. (D) Violin plots showing the distributions of $\log_2$ transformed transcription efficiency (TE, y-axis) for each GTF (x-axis). TE indicates the number of mRNA molecules synthesized during one binding event. The points show the medians of the $\log_2$ transformed TE values. Mean and median TE values are shown above the plots. Number of observations (*n*): TBP = 1572, TFIIA = 1382, TFIIB = 1975, TFIIF = 1560, TFIIE = 2163. (E) PCA plot showing a low-dimensional representation of gene targets based on GTF residence times. Each gray point is a gene, color map shows the mean synthesis rate of genes under a given area. The percentage within the axis labels indicates the percentage of variance explained by a given PC. (F) Pearson's correlation coefficients (y-axis) between the indicated PCs (panel title) and GTF (x-axis) residence times. (G) Pearson's correlation coefficients (y-axis) between PCs (x-axis) and synthesis rates. Source data are available online for this figure.

We next looked for pairwise relationships between the chromatin binding residence times of each GTF, and highlighted each gene by transcription rate (Fig. EV4). This allowed us to identify a cluster of highly transcribed genes (dark cluster) associated with the presence of long-lived TFIIE and TFIIF, as well as TFIIB residence times that were in a similar range of several minutes. This was in contrast to TBP and TFIIA, whose residence times did not show any significant pattern. TBP was overall less informative as most TBP binding events measured were short-lived and not well correlated with transcription rate (Fig. EV3C). In fact, the residence times of TFIIA, TFIIB, and TFIIF were individually not correlated with transcription rate either. This was in contrast to

the positive correlation that was observed between TFIIE residence time and transcription rate (Fig. EV3).

Next, we clustered all the genes for which we obtained residence time measurements for four factors (TFIIA, TFIIB, TFIIE, and TFIIF ($n = 1417$)). We omitted TBP from this analysis due to the reduced number of sites with reliably estimated residence times >1 min. We identified ten clusters spanning the full range of transcription rates (Fig. 4A,B; Dataset EV3). Consistent with the results presented above, the most highly expressed genes had longer-lived TFIIE and/or TFIIF, whereas poorly expressed genes had promoters with longer-lived TFIIA. Longer residence times of TFIIB were associated with genes in several clusters, including particular genes that were poorly expressed (cluster 8; Fig. 4A,B). Notably, the relatively long residence times of both TFIIE and TFIIF at cluster 1 promoters were associated with the production of multiple mRNAs, suggesting the formation of stable (sub) complexes that promote transcriptional bursting.

In support of the biological significance of the observed residence time differences, genes within clusters 1–7 were functionally related (Fig. 4C). Cluster 1 genes include ribosomal protein genes and genes involved in RNA binding and translation. Additionally, cluster 2 genes are involved in biosynthetic processes; cluster 3 genes include those involved in Golgi organization; cluster 4 genes are involved in localization, transport, and the proteasome; cluster 5 genes are involved in proteasome degradation; cluster 6 genes have roles in nucleocytoplasmic transport; and cluster 7 genes are involved in proteasome and protein–lipid complex organization. The longer GTF residence times (as well as higher gene expression rates) at ribosomal protein genes in cluster 1 compared to the GTF residence times at other genes are statistically highly significant (Fig. EV5A–C, cluster 1 gene CC signal tracks shown in Appendix Fig. S3). Moreover, the expression of genes in most of these clusters is controlled by particular TFs (or sets of TFs; Fig. 4D,E; Appendix Fig. S4), suggesting a mechanistic relationship between particular TFs and PIC assembly dynamics. Modest but significant increases in TFIIA and TFIIE residence times were observed at promoters with strong TATA elements versus those without such an element (Basehoar et al, 2004); these changes were consistent with a significant increase in RNA synthesis rate driven by TATA-containing promoters versus those without strong TATA elements (Appendix Fig. S5A–C). In contrast, we did not observe any significant differences between the residence times of galactose-induced genes and all other genes, even though the synthesis rates of the *GAL* genes were significantly higher than the genes that were not induced by galactose (Appendix Fig. S6A, B).

An unexpected observation was the abovementioned bimodal distribution of TFIIF residence times (Figs. 2F and EV4). We observed functional enrichment of the genes in each of these two classes, with promoters in both classes associated with different subsets of genes involved in translation/ribosome. Genes with short-lived TFIIF were further associated with other biosynthetic processes (Appendix Fig. S7A,B). Consistent with this, the genes in the long-lived and short-lived TFIIF classes were associated with particular enriched TFs, some of which were shared (Appendix Fig. S7C–F). Among the TFs associated with genes in the long- and short-lived TFIIF classes, Rap1 was of particular interest as competition ChIP data were available for Rap1 from a prior study (Lickwar et al, 2012). Although Rap1 residence times were not correlated with residence times for TBP or TFIIB, there was a

moderate correlation between Rap1 residence times and the residence times for TFIIA and TFIIE (Pearson's correlation coefficients ~0.37 and 0.3, respectively), and Rap1 residence times were significantly longer at genes with long-lived TFIIF compared to genes with short-lived TFIIF (Appendix Fig. S8A,B).

## Discussion

The computational approach employed here for the extraction of kinetic parameters from CC data is well supported by comparison with previous work. The TBP residence times obtained by analysis of CC data in this study were correlated with the residence times obtained from an older study using microarray data (Appendix Fig. S8C; (van Werven et al, 2009)) and are also broadly consistent with kinetic results for TBP in human cells (Hasegawa and Struhl, 2019). This includes the rank order in which tRNA genes had much longer residence times than mRNA genes. Previously, we used a formaldehyde crosslinking kinetic approach, called CLK, to measure chromatin binding dynamics (Poorey et al, 2013). While the CLK method is technically challenging as well as locus-specific (Zaidi et al, 2017b), we observed a rough agreement between the kinetic parameters obtained by the two methods for the handful of loci for which complementary measurements are available (Appendix Fig. S8D).

Live cell imaging has revealed that the majority of TF-chromatin interactions studied are short-lived, with residence times on the order of seconds (van Royen et al, 2011; Paakinaho et al, 2017; Normanno et al, 2012; Swinstead et al, 2016; Liu and Tjian, 2018; Lionnet and Wu, 2021; Brouwer and Lenstra, 2019; Nguyen et al, 2021). This includes TFIIB (Zhang et al, 2016; Nguyen et al, 2021; Sprouse et al, 2008), for which CC results are reported here. The observation of highly dynamic binding by TFs has led to the view that such dynamics enable temporally responsive regulation of gene expression, and that TF residence times are associated with the duration of bursts in which more than one RNA molecule is synthesized during the TF period of occupancy on the promoter (Donovan et al, 2019; Coulon et al, 2013; Nicolas et al, 2017; Lenstra et al, 2016). Consistent with the observation of frequent short-lived chromatin interactions for TFs, we observed that the majority of the interactions between TBP, TFIIA, TFIIB, or TFIIF and chromatin had residence times of less than 1 min (Fig. 2F). It was not possible to reliably estimate the residence times of these short-lived interactions using CC, but they must last long enough to be captured by crosslinking. It is likely that other very short-lived interactions were not detectable by our method because of their inability to be crosslinked. Conversely, it is possible that long-lived chromatin interactions such as those we report here would be difficult to detect with live cell imaging, particularly if they occur infrequently, although evidence is emerging for TF-chromatin binding residence times on the minutes time scale using live cell imaging (Hipp et al, 2019). Since the formation of a PIC is mutually dependent on all of the GTFs (Petrenko et al, 2019) and structural data are consistent with the requirement for, e.g., TBP and TFIIB binding to establish a platform for the binding of Pol II, TFIIF and TFIIE (Osman and Cramer, 2020; Nogales et al, 2017), it may appear counterintuitive that such "early" binding GTFs have shorter residence times than the "late" binding GTFs at many promoters. It is important to recognize that the residence times that

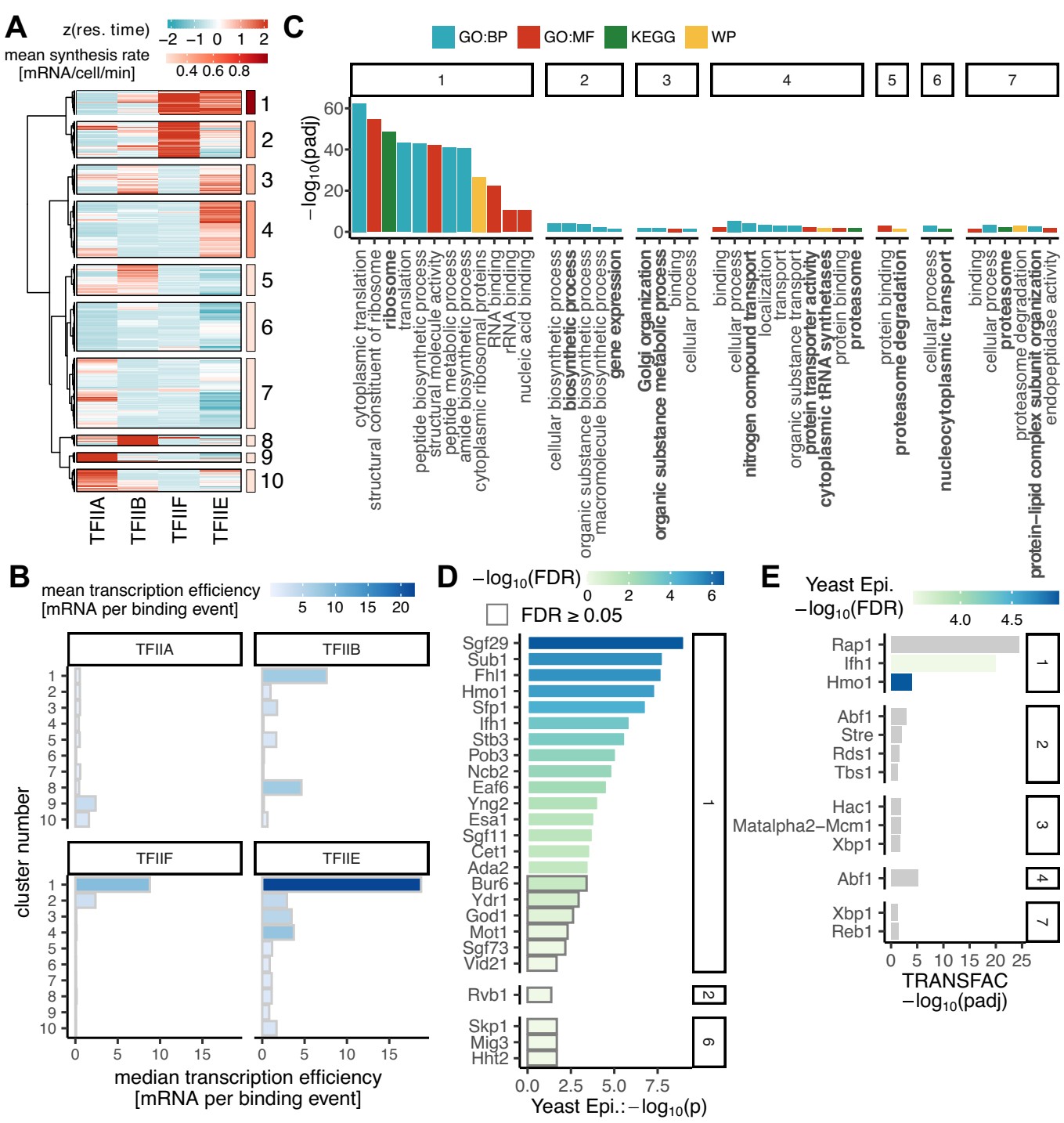

**Figure 4.  Gene classes based on GTF residence times combinations.**

(A) Heatmap showing z-score normalized residence times of the indicated GTFs (columns) across gene targets (rows) with the available residence time estimations from all four GTFs (*n* = 1417). Colored panels on the right side show the mean synthesis rates of genes belonging to the ten clusters. (B) Bar plots showing median TE (x-axis) within clusters (y-axis) from (A) and color-coded based on mean TE. (C–E) Functional annotation of genes from clusters in (A). The cluster number is indicated in the panel titles. (C) Pathway enrichment. *P*adj < 0.05. (D) Yeast Epigenome database DBF enrichment excluding subunits of GTFs and Pol II. Fisher's *P* < 0.05. (E) TRANSFAC enrichment. Fisher's *P*adj < 0.05. Colored bars were identified as significantly enriched (FDR <0.05) in the Yeast Epigenome database.

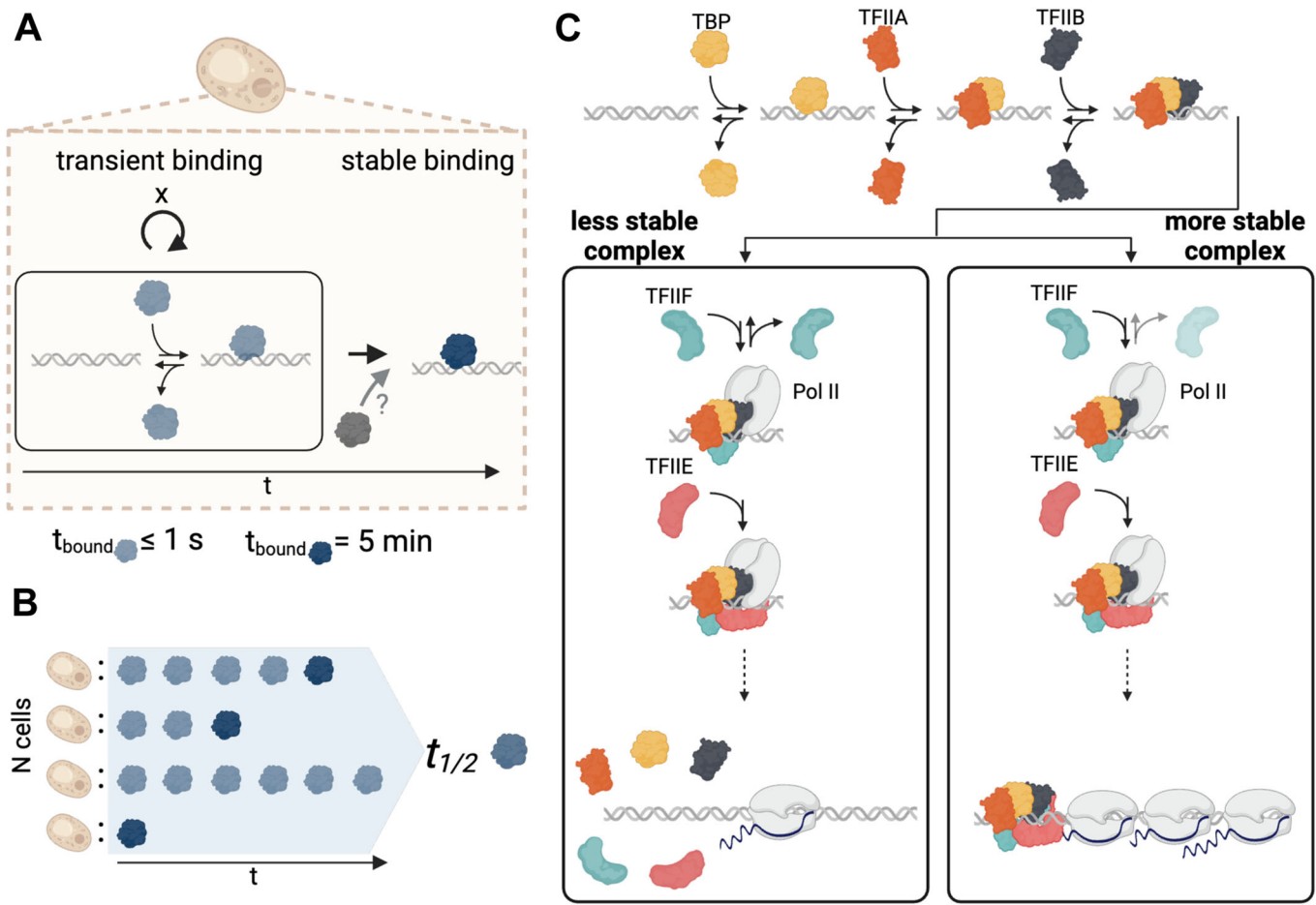

**Figure 5. Model.**

(A,B) Interpretation of residence times. (**A**) A GTF can undergo multiple rounds of transient binding (indicated by x and shown in light blue) before it binds stably (dark blue), possibly assisted by the other factors (gray). (**B**) The final residence time estimates at a given site ($t_{1/2}$, shaded blue) represent the average of transient and stable binding over the course of the experiment across all cells. Multiple transient binding events over time are shown in light blue; stable binding events block sites from exchange and are shown in dark blue. (**C**) The results suggest that for the majority of genes, PICs are unstable until TFIIE binding, which leads to functional PIC assembly, the initiation of RNA synthesis, and the release of Pol II and PIC disassembly. At a relatively small subset of genes e.g., genes coding for ribosomal subunits, relatively stable PICs are formed upon TFIIF binding (note lighter color for disassociation) and further stabilized by TFIIE binding, followed by the initiation of RNA synthesis. Upon Pol II release, stable PICs may be disassembled or at certain promoters may be stable and lead to transcriptional bursting. The formation of more stable PICs is likely associated with promoter-specific features and cofactors. The figure is meant to be illustrative and does not represent accurate sizes or molecular shapes of the factors of interest.

we report here reflect the global average of the residence times for all complexes formed in vivo that contain the GTF of interest (Fig. 5A,B). Thus, core promoter-bound complexes that contain TFIIE or TFIIF are highly likely to contain TBP and TFIIB. We infer that the reason there is shorter-lived TBP and TFIIB at many promoters with longer-lived TFIIF and TFIIE is that most TBP- and TFIIB-containing complexes do not lead to the formation of a complex of a productive PIC that contains TFIIF or TFIIE (Fig. 5C). These considerations, in addition to the focus in the present study on promoter regions, could explain, at least in part, why TFIIE was observed by single-molecule tracking to be engaged primarily in short-lived binding events whereas long-lived TBP binding events were observed (Nguyen et al, 2021). In this regard, it is worth noting that the highly dynamic behavior of TBP reported here is consistent with highly mobile TBP in the nucleoplasm overall, which was dependent on the TBP-DNA dissociating enzyme Mot1

and observed in live cells by fluorescence recovery after photobleaching (FRAP, ref. (Sprouse et al, 2008)).

The biological significance of the residence times reported here is supported by the functional enrichment of genes in each of the clusters (Fig. 4C). This argues strongly that GTF residence time dynamics are tuned to facilitate expression levels that ensure that cells function and respond in physiologically appropriate ways. Since these gene sets are controlled by specific sets of TFs (Fig. 4D,E), it is reasonable to suggest that GTF dynamics are influenced in predictable ways by the TFs that control the expression of the associated genes. It is understood that TFs exert context-specific effects on gene expression, and such effects have been generally described in terms of effects mediated by coregulatory interactions with other TFs as well as epigenetic control, including DNA methylation (Stone et al, 2019; Mony et al, 2021; Fertig et al, 2013). In future work, it could be interesting to explore

how GTF residence times are impacted by the manipulation of such regulators. We suggest that RNA output resulting from the interplay of these variables is at least partly a consequence of the capacity to catalyze the formation of functional PICs by overcoming kinetic bottlenecks in PIC assembly that are also related to the underlying DNA sequence and chromatin environment.

A striking observation from the results of this study is that the residence time of TFIIE is correlated with the mRNA synthesis rate, and the number of mRNA molecules produced during a TFIIE residence time suggests that one TFIIE binding event is associated with the production of one mRNA molecule (Fig. 3D). This is in contrast to the other GTFs for which one binding event was associated with less than one mRNA molecule produced. We were not able to measure Pol II directly using CC because we do not have a system for inducing the expression of all of the Pol II subunits to generate a competitor isoform of Pol II. However, TFIIF can serve as a proxy for Pol II itself as biochemical and structural data support a model in which TFIIF can enter the PIC in association with Pol II (Bushnell et al, 1996; Ranish and Hahn, 1996; Orphanides et al, 1996; Sainsbury et al, 2015; Osman and Cramer, 2020). The combined results suggest that the formation of a PIC is an inefficient process in vivo, with most interactions of GTFs leading to subcomplexes that decay rather than leading to the formation of a PIC capable of producing mRNA. This general view of transcription initiation inefficiency is consistent with live cell imaging data obtained by analysis of a gene array in a mouse cell line (Stasevich et al, 2014). Moreover, this pattern is broadly consistent with a PIC assembly pathway derived from in vitro studies in which TBP/TFIID initially interacts with DNA directly, followed by the binding of TFIIA and TFIIB, which provide a platform for the binding of Pol II and TFIIF, and subsequently TFIIE (Fig. 5C; (Luse, 2014)). The broad outlines of the pathway suggested here are compatible with the notion of dynamic and even branching assembly pathways proposed on the basis of observations of single complexes formed using nuclear extracts (Baek et al, 2021). As the work presented here includes a genome-scale inventory of kinetic behavior and most promoters do not possess a regulatory region (Rossi et al, 2021), the distribution of residence times reflects the behavior of GTFs at such promoters. This probably explains why, for example, we observe promoters with a wide range of TFIIE residence times (including promoters where it is relatively long-lived), whereas Baek et al (2021) observed unexpectedly dynamically bound TFIIE in that system. Additionally, our data do not have sufficient resolution to distinguish GTF loading at regulatory regions (Baek et al, 2021) versus core promoters, and we are therefore unable to draw inferences about the impact of activators per se on the assembly process. We infer the existence of stable TFIIB complexes on the basis of slow turnover at a relatively small number of genes; it appears that most TFIIB-containing complexes are unstable and that assembly of TFIIB in the PIC requires Pol II (Nguyen et al, 2021). Despite the dispensability of some GTFs in vitro under certain conditions, our results are also consistent with depletion experiments showing that all of the GTFs are required for all Pol II-mediated transcription in vivo, and that stable, partially assembled PICs are not detectable (Petrenko et al, 2019). Of note, however, we did observe a small number of relatively long-lived complexes containing TFIIA or TFIIB (Fig. 2D,F; Dataset EV1). Such long-lived complexes could be consistent with the formation of a subcomplex of GTFs that is

durably bound to promoters and promotes reinitiation (Yudkovsky et al, 2000). The formation of long-lived scaffolds of GTFs at some promoters is also suggested by the residence times of TFIIE and TFIIF at Cluster 1 genes, which were associated with the production of multiple mRNAs (Fig. 4B). Lastly, our analysis includes the minimal set of GTFs required for in vitro transcription using a naked DNA template (Tyree et al, 1993; Fujiwara and Murakami, 2019; Luse, 2019). Given the complexity in interpreting the results from a CC experiment in which one overexpressed a single or a few subunits of a multisubunit complex, it could be difficult to apply this method to the analysis of multisubunit complexes as currently implemented. In future work and using methods suitable for the analysis of multisubunit complexes, it will be interesting to investigate the dynamics of TFIID, TFIIH (Greber et al, 2019; Nogales and Greber, 2019), Mediator, and Pol II itself (Nozawa et al, 2017; Plaschka et al, 2015). Other important questions that could be addressed by performing kinetic measurements in suitably perturbed cells include probing the roles of promoter chromatin structure, particularly the function of the first nucleosome (Petrenko et al, 2019). Taken together, we feel that the results presented here provide a foundation for future work to understand how TFs, cofactors, and the native chromatin environment contribute mechanistically to the establishment of the rates of transcription initiation observed in vivo.

## Conclusions

The results reported here provide a wealth of kinetic information describing the chromatin binding dynamics of five key GTFs at the majority of promoters in budding yeast. In general agreement with live cell imaging results, we find that many interactions are too short-lived to be measured by CC. However, there are many interactions with residence times in the several-minute range, and importantly, promoters with shared GTF kinetics are functionally related. This supports a model in which the rates of RNA synthesis in vivo are influenced or perhaps controlled by rates of PIC assembly, which themselves result from the combination of promoter sequence, chromatin environment, and the TFs and cofactors that impact them. Overall, the kinetic behavior is consistent with the stepwise PIC assembly pathway established using purified components in vitro in which the RNA synthesis rate is closely correlated with the residence time of TFIIE. These results suggest that at most promoters, relatively unstable GTF subcomplexes give rise to more stable fully assembled PICs and that the initiation of RNA synthesis is accompanied by PIC dissolution. At certain promoters, GTF binding events are associated with the production of multiple mRNAs, suggesting the formation of stable PIC subcomplexes that facilitate transcription reinitiation.

## Methods

### Yeast strains

The parental diploid strain W303 (Ralser et al, 2012) was used to generate all of the competition ChIP strains. For each GTF, one allele was N-terminally tagged with 3xHA and placed under the control of an inducible *GAL1* promoter. The other allele was

N-terminally tagged with 9xMyc and remained under the control of the endogenous promoter (Longtine et al, 1998). For the measurement of GTFs with two subunits, one allele of each subunit was placed under *GAL1* control, and one subunit was tagged with Myc or HA. For TFIIA, the Toa1 subunit was epitope-tagged; for TFIIE, the Tfa1 subunit was tagged, and for TFIIF, the Tfg2 subunit was tagged.

For construction of the *GAL1*-induced alleles, the plasmid pFA6-His3MX6-PGAL1-3HA (RRID:Addgene_41610, ref. (Longtine et al, 1998)) was used to obtain the His3MX6-PGAL1-3HA cassette by PCR amplification (see Appendix Table S2 for primers) and was integrated into the genome using standard yeast molecular biology techniques. For the GTFs TFIIA, TFIIE, and TFIIF, which consist of two subunits, one copy of each subunit was placed under *GAL1* control to ensure balanced expression of the competitor isoform. Following the integration of the *HIS3-GAL1*-3HA cassette at one gene subunit, the strain was transformed with the *TRP1-GAL1* cassette from pFA6-TRP1-PGAL1 (RRID:Addgene_41606, ref. (Longtine et al, 1998)), placing the second subunit under *GAL1* control but without an epitope tag. The 9xMyc tag was integrated into the genome of another isolate of W303 using the integration and Cre-recombinase knockout method and reagents developed by Gauss et al (Gauss et al, 2005). The 9xMyc tag and loxP-flanked KanMX6 marker were PCR amplified from pOM20 and integrated into the yeast genome using standard methods as above. The KanMX6 marker was then then deleted using the *GAL*-inducible Cre-recombinase carried on the plasmid pSH47 (Güldener et al, 1996). The Myc-tagged strains were then transformed with pRS319 (RRID:Addgene_35459, ref. (Sikorski and Hieter, 1989)) to introduce a *LEU3* marker for selection. In subsequent steps, diploid strains with HA- or Myc-tagged alleles were sporulated, and haploid segregants were mated to yield the competition ChIP (CC) strains with different tags on each of the alleles for the GTF of interest. Proper integration and function of the targeted alleles were confirmed for all strains by PCR (Appendix Table S2 for primers), Western blotting using anti-HA or anti-Myc antibodies, and targeted DNA sequencing of the modified loci. Functionality of the Myc- and HA-tagged alleles was confirmed by spot tests and streaking of the strains on appropriate media (Appendix Fig. S9).

## Western blotting

To measure the time course of synthesis of the *GAL1*-induced alleles, CC strains were grown in 175 ml YEP + 2% raffinose. At OD600 of 0.6, a 20 ml aliquot of the culture was collected for the 0 min time point and 11 ml of 30% galactose was added to the remaining culture. About 20 ml aliquots were removed at 10, 20, 25, 30, 40, 60, 90, and 120 min after galactose addition, and whole-cell extracts were prepared from them as described previously (Zaidi et al, 2017b). Whole-cell extract protein was resolved on 10–12% SDS-Page gels (depending on the size of the tagged protein). The protein was transferred overnight to 0.22 μ PVDF membranes and probed using either anti-HA (Abcam Cat# ab9110, RRID:AB_307019) or anti-Myc (Abcam Cat# ab32, RRID:AB_303599) antibodies followed by detection using either the HRP-conjugated goat anti-mouse secondary antibody, (for Myc; Thermo Fisher Scientific Cat# 31430, RRID:AB_228307) or goat anti-rabbit secondary antibody (for HA; Thermo Fisher Scientific Cat# 31460, RRID:AB_228341) and ECL substrate (Thermo Fisher

Scientific Cat# 32106). Although the galactose-induction experiments were conducted in the same way for each strain, for unknown reasons, we observed reproducible differences in the time course of accumulation of the competitor (Fig. 2C). Since the promoter is the same for each factor, we presume this reflects differences in the rate of protein synthesis in vivo post-transcription. The CC method relies on measuring a difference between the rate of HA/Myc isoform turnover versus the rate of synthesis of the competitor (HA-isoform), and for this reason, factor-specific differences in the rate of competitor synthesis do not impact the results reported here.

## CC time course experiments and ChIP-seq library preparation

Each CC strain was inoculated in 100 ml YEP + 2% raffinose at 30 °C and incubated overnight. These starter cultures were then used the next day to inoculate 2250 ml cultures of YEP + 2% raffinose at an initial OD600 of 0.05. When an OD of 0.6 was reached, for the 0-min time point, 250 ml of the culture was crosslinked by adding 6.75 ml formaldehyde (Thermo Fisher Scientific Cat# F79-500) to achieve a final concentration of 1% for 20 min. The reaction was then quenched by adding 15 ml of 2.5 M glycine for 5 min, and the cells were collected by centrifugation. To the rest of the 2000 ml culture, 142.8 ml of 30% galactose was added to yield a final concentration of 2%. At 10, 20, 25, 30, 40, 60, 90, and 120-min time points, 250 ml of the culture was collected, crosslinked, and quenched the same way as the 0-min time point. Cell pellets were washed three times with TBS buffer (40 mM Tris-HCl, pH 7.5 plus 300 mM NaCl), and ChIP was performed as described (Viswanathan et al, 2014). The HA and Myc antibodies used for ChIP were the same as those used for western blotting described above. Successful ChIP was confirmed by RT-PCR using primers to detect binding to the *URA3* promoter (5′-AAGATGCCCATCACCAAAA-3′ and 5′-AAGAATACCGGTTCCCGATG-3′). ChIP-seq libraries were prepared following the manufacturer's instructions using the Illumina TruSeq ChIP library prep kit set A and B (Cat# IP-202-1012 and IP-202-1024). Successful amplification was confirmed by RT-PCR using the *URA3* promoter primers. Library quality was assessed using an Agilent Bioanalyzer 2100 and the Agilent-1000 DNA kit (Agilent Cat# 5067-1504), and libraries were quantified using the Qubit dsDNA Quantitation, High Sensitivity kit (Cat# Q32851). A 5 nM pool of each library was sequenced by the UVA Genome Analysis and Technology Core (RRID:SCR_018883) using Illumina Next-Seq500 and NextSeq2000 instruments. While we only gathered one replicate competition ChIP-seq sample per time point, the normalized data from the eight-time points were required to fit both the Hill model and turnover model with $R^2 > 0.7$, as detailed below.

## Nascent RNA labeling

Nascent RNA labeling was performed as previously described (Warfield et al, 2017). Briefly, W303 cells were grown as for competition ChIP and induced with 2% galactose for 20 or 60 min. An 800 ml culture in YEP + 2% raffinose was grown at 30 °C to an OD600 of 0.6, then 57 ml of 30% galactose was added. Twenty minutes after galactose addition, 400 ml of the culture was divided

into 200 ml aliquots, and 500 µl of 2 M 4-thiouracil (4-sU, Sigma-Aldrich Cat# 440736-1 G) was added to one of the flasks with vigorous mixing and returned to the shaking incubator for 6 min. Cells with and without 4-sU were pelleted and washed with TBS. At the 60-min time point, the remaining 400 ml culture was split and treated as described for the 20-min time point culture. Two biological replicates were obtained for each condition.

*S. pombe* strain SY78 cells were used as a spike-in normalization control. About 100 ml of *S. pombe* cells were grown in YE media (0.5% yeast extract plus 3% glucose) to an OD600 of 0.6 and labeled by adding 125 µl of 2 M 4-sU for 6 min and collected by centrifugation.

The *S. cerevisiae* W303 cells and *S. pombe* SY78 cells were mixed in an 8:1 ratio for each condition, and RNA was isolated using the Ribopure Yeast Kit (Ambion Cat# AM1924). About 40 µg of RNA was biotinylated with 4 µg of MTSEA Biotin XX (Biotium Cat# 90066). The biotinylated RNA was isolated by binding to 80 µl of a Dynabeads MyOne Streptavidin C1 bead suspension (Invitrogen Cat# 65001) by rotating the tube for 15 min, and the unbound supernatant was saved. The bound RNA was eluted in 50 µl of streptavidin elution buffer. The eluted RNA and the RNA in the flowthrough were purified and concentrated using RNeasy columns (Qiagen Cat# 74104).

## RNA-seq

Ribosomal RNA was depleted using the Ribo Minus Yeast module (Thermo Fisher Scientific Cat# 45-7013) and libraries were constructed using the Ultra Directional RNA Library Prep Kit (NEBNext Cat# E74205) and Multiplex Oligos (NEBNext Cat# E73355). Sequencing was performed by Novogene using the Illumina NovaSeq 6000 platform.

## Preprocessing of high throughput DNA sequencing data

Libraries prepared from each time point for a given GTF and for either HA- or Myc-tagged samples were sequenced in a single multiplexed run. Raw read quality was assessed using FASTQC (v0.11.5) (Andrews, 2010). Fastq files from individual flow cells were merged, and reads were mapped to the sacSer3 reference genome using Bowtie2 (v2.2.6) (Langmead and Salzberg, 2012) with default settings. Overall read mapping was typically in the 90+ % range, yielding ~20–30 M reads per time point on average. The resulting SAM files were converted to BAM format, unmapped reads were removed, and the BAM files were sorted and indexed using SAMtools (v0.1.19-44428 cd) (Li et al, 2009). The landscape of read mapping was inspected using the Integrated Genomics Viewer (IGV) (Thorvaldsdottir et al, 2013), and peaks of enrichment were identified using MACS2 (v2.1.0.20151222) (Zhang et al, 2008) applied to each of several early time point Myc datasets with an input dataset as control and options --nomodel --extsize 147. Peaks from individual MACS2 runs were browsed in IGV, then concatenated and merged using the bedtools (v2.18.2) *merge* function (Quinlan and Hall, 2010). Count tables were then generated by associating reads with the peak intervals using bedtools *multicov*. Read counts were normalized in a three-step process. First, read counts in each peak and for each time point were normalized to the overall read depth. Next, read counts for the HA samples were normalized to the average relative levels of the factor of interest using the average values obtained from three independent western blots. Lastly, the normalized HA read count matrix was divided by the normalized Myc count matrix to yield the ratio count tables for mathematical modeling as described below. Importantly, this normalization approach was validated by comparison with earlier results: residence times derived from normalized TBP CC data were strikingly well correlated with TBP CC data obtained many years earlier and using arrays rather than sequencing (Appendix Fig. S8C). Since most GTF binding events display fast, second-timescale dynamics (Nguyen et al, 2021), this normalization method tends to bring the kinetics of fast sites in line with the timescale of competitor induction regardless of whether there is a delay in e.g., the time it takes for competitor protein maturation or nuclear import. Notably, the turnover model assumes the protein induction is effectively occurring in the nucleus, which this normalization approximates.

## Deriving residence times from competition ChIP-seq ratio data using a mass action kinetics turnover model

We adapted the approach of Zaidi et al, (Zaidi et al, 2017a), originally developed for TBP competition ChIP-chip data, to fit a differential equation-based turnover model at every GTF site using normalized competition ChIP-seq data from multiple GTFs. We used normalized count tables (see previous section of Materials and Methods) with HA/Myc ratios for every GTF site, $R(t)$, for every time point, $t$. We ultimately estimate the ratio of fractional occupancies of HA- over Myc-tagged GTF, $\theta_B(t)/\theta_A(t)$ with $B$ and $A$ representing HA- and Myc-tagged proteins, respectively, from $R(t)$ at every time point. We then fit a mass action kinetic turnover model to the estimated ratio of fractional occupancies at every promoter site where a peak was identified to derive the residence time for a GTF at that site. As previously reported (Lickwar et al, 2012; Zaidi et al, 2017a) and detailed below, the ratio, $\theta_B(t)/\theta_A(t)$, is insensitive to the overall on-rate, which is the only place that the concentrations of HA- and Myc-tagged proteins enters the mass action model. Consequently, the estimation of residence time is insensitive to the relative levels of HA- and Myc-tagged proteins at steady state.

More specifically, we first fit the normalized ratio of HA- over Myc-tagged relative protein levels as estimated by Western blotting versus induction time, which we denote $c_B(t)/c_A$ with $B$ and $A$ representing HA- and Myc-tagged protein, respectively, to a Hill model

$$c_B(t)/c_A = X_P\left(\left(t/t_{1/2ind}\right)^n / \left(1 + \left(t/t_{1/2ind}\right)^n\right)\right) \qquad (1)$$

In Appendix Table S3, we show the resulting fitted parameters ($X_P$, $t_{1/2ind}$) and statistics associated with the significance of each parameter's contribution to the fit for every GTF. In this case, we fixed the Hill coefficient, $n$, to be an integer and selected the value that maximized the adjusted $R^2$. In order to satisfy the $t = 0$ and $t \to \infty$ boundary condition of the mass action kinetic turnover model shown below in Eqs. (2) and (3), which are $\theta_B(0)/\theta_A(0) = 0$ and $\lim_{t \to \infty} \theta_B(t)/\theta_A(t) = X_P$, we subtract the residual background and scale the normalized competition ChIP-seq ratio data at every site where peaks were called as follows. We fit the data to a Hill model with the form shown in Eq. (1) with the same $n$ and an added background variable $B$ at every site. This yields an amplitude,

$X_{CC}$, a half-time rise, $t_{1/2CC}$, and background $B$ for every site. We estimate the ratio of HA- over Myc-tagged GTF occupancy, $\theta_B(t)/\theta_A(t)$, at every site for every time point, $t$, by subtracting the residual background $B$ from the normalized ChIP signal ratio data, $R(t)$, and scaling the result: $\theta_B(t)/\theta_A(t) = (X_P/X_{CC})(R(t) - B)$. We then effectively solve the following coupled differential equations, which model each GTF's turnover at every site which we assume follows mass action kinetics, where $k_a$ and $k_d$ are the molecular on- and off-rate, respectively:

$$\frac{d\theta_B(t)}{dt} = (k_a c_A)\frac{c_B(t)}{c_A}(1 - \theta_A(t) - \theta_B(t)) - k_d\theta_B(t) \quad (2)$$

$$\frac{d\theta_A(t)}{dt} = (k_a c_A)(1 - \theta_A(t) - \theta_B(t)) - k_d\theta_A(t) \quad (3)$$

We assume that these rates are the same for both HA- and Myc-tagged GTFs. These coupled equations cannot be solved analytically. Thus, we effectively solve them and fit the resulting ratio of occupancies, $\theta_B(t)/\theta_A(t)$, to the background subtracted, scaled competition ChIP-seq data using Mathematica. Briefly, we use the function *ParametricNDSolveValue* twice to return an effective, numerical solution of Eqs. (2) and (3) as a function of the parameters $k_a c_A$ and $k_d$: $\theta_B(t; k_a c_A, k_d)$ and $\theta_A(t; k_a c_A, k_d)$. We then take the ratio of the outputs of ParametricNDSolveValue, $\theta_B(t; k_a c_A, k_d)/\theta_A(t; k_a c_A, k_d)$, and input it into *NonlinearModelFit* which then fits this ratio to the background subtracted, scaled competition ChIP-seq data. We and others formally show the ratio of fractional occupancies is relatively insensitive to the on-rate, $k_a c_A$, while being highly sensitive to the off-rate, $k_d$. We derive the physical residence time for every GTF at every site using $t_{1/2} = \ln2/k_d$. Finally, we make use of an observation made in (Zaidi et al, 2017a) to make precise starting estimates of the residence time for non-linear model fitting using *NonlinearModelFit*. Specifically, the residence time is well approximated by a relatively simple linear or quadratic function of $t_{1/2CC} - t_{1/2ind}$ derived by fitting a Hill model to the normalized competition ChIP-seq ratio data at every site and the ratio of GTF protein levels as a function of time. We start with an initial guess that works well for most GTFs: $t_{1/2}^0 = 0.6(t_{1/2CC} - t_{1/2ind}) + 0.1$ (Fig. 1D), perform the fit of the actual turnover model to the scaled, background subtracted competition ChIP-seq data, derive estimates of $t_{1/2}$, fit $t_{1/2}$ to linear or quadratic functions of $t_{1/2CC} - t_{1/2ind}$, use this more precise relationship of an initial estimate of residence time, $t_{1/2}^0$, and refit the turnover model to the competition ChIP data. In Appendix Table S4, we show the initialization formulas used for the final turnover model fit the competition ChIP-seq data used to derive the final estimates of residence times for every GTF. Finally, *NonlinerModelFit* returns a number of statistics associated with the fit at every site. This includes an error estimate of the off-rate, $\triangle k_d$, and the adjusted $R^2$. Sites that yielded a relative error $\triangle k_d/k_d < 3$ and adjusted $R^2 > 0.7$ were used in downstream analysis involving residence time estimates.

## Fitting additional reliably fast sites

After the initial fitting, additional reliably fast sites were added to the estimated residence times. These were identified by fitting Hill equation Eq. (1) with the R *nls* function to the normalized HA/Myc count ratios which were further normalized to range between zero and one. Hill coefficients were provided from protein induction curve fits (Figs. 2C and EV1E). Initial estimates for fitting the Hill model using the *nls* function were set with parameter $start = list(t_{1/2CC} = 40, X_{CC} = 1)$, and parameter *control* was set to *nlc*. For each GTF, sites without estimated residence times from the turnover model whose $\triangle t_{1/2} = t_{1/2CC} - t_{1/2ind}$ (Fig. 1D) were less than 2 min were classified as reliably fast (<1 min). All residence time estimates are available in Dataset EV1.

For plotting purposes, the residence times for the reliably fast sites were generated with the R *runif* function with $min = 0$, $max = 1$. At the beginning of each script, the function *set.seed* was used with parameter *42* for reproducibility. In each plot, the randomly generated values are highlighted either by their separation by a dashed line or shaded area.

## Gene assignment and filtering

Individual regions were assigned to the nearest genes with *calcFeatureDist_aY* function (available from https://github.com/AubleLab/annotateYeast) with default parameters. Only regions within −250 to 100 bp from transcription start sites (TSSs) were kept. If multiple regions were assigned to one gene, only the closest one was kept. Regions assigned to tRNAs were removed from the analysis.

## Nascent RNA-seq analysis

Raw paired-end FASTQ files were mapped to the *S. cerevisiae* genome (http://daehwankimlab.github.io/hisat2/download/#s-cerevisiae, R64-1-1) with HISAT2 (2.0.4) (Kim et al, 2019) with parameter *--rna-strandness RF* and converted to BAM files using SAMtools (0.1.19-44428 cd) (Li et al, 2009) *view* function with parameters *-S -b*. SAMtools *sort* and *index* functions with default parameters were used to sort and index the BAM alignment files.

To create alignment indexes for *S. pombe* (used for normalization), the *S. pombe* FASTA file (ASM294v2) was obtained from Ensembl (Cunningham et al, 2022) and converted to an index file with the *hisat2-build* function with default parameters. The paired-end FASTQ files were then mapped against the created index files and further processed analogously to *S. cerevisiae*.

The quality of both FASTQ and BAM files was assessed with FastQC (0.11.5) (Andrews, 2010) in combination with multiQC (v1.11) (Ewels et al, 2016), and BAM files were further visually inspected with IGV (2.7.2) (Thorvaldsdottir et al, 2013).

The aligned reads were quantified over *S. cerevisiae* genes using Rsubread (2.4.3) (Liao et al, 2014) *featureCounts* function with parameters *GTF.featureType = "gene"*, *GTF.attrType = "gene_id"*, *countMultiMappingReads = TRUE*, *strandSpecific = 2*, *isPairedEnd = TRUE*. The GTF and FASTA files provided to the function were obtained from Ensembl (Cunningham et al, 2022), genome assembly R64-1-1. To normalize the data, normalization factors for each sample were calculated as the total number of reads mapped to *S. pombe* divided by 2,000,000. The normalized counts were obtained by dividing the raw counts by each sample's corresponding normalization factor. Genes with 0 counts in more than half of the samples were filtered out.

Principal component analysis (PCA) was performed by first creating a *DESeq* object from the raw count table (with low count genes filtered out) with the DESeq2 (1.30.1) (Love et al, 2014)

*DESeqDataSetFromMatrix* function followed by *S. pombe* normalization with DESeq2 *normalizationFactors* and regularized log transformation with DESeq2 *rlog* function with parameter *blind* = *TRUE*. The resulting object was passed to R *prcomp* function.

DESeq2 was used to identify any differences in gene expression between samples grown for 20 or 60 min in galactose. Raw counts from samples with thiouracil addition were passed to *DESeqDataSetFromMatrix* function with *design* parameter set to *time in galactose*. S. pombe normalization factors were set with *normalizationFactors*. Genes with adjusted *p* value ($p_{adj}$) <0.05 were considered differentially expressed between the two conditions.

Synthesis rates were estimated with DTA (2.36.0) (Schwalb et al, 2012) *DTA.estimate* function. *S. pombe*-normalized counts from samples with thiouracil addition were used for the analysis. All genes with 0 count in any of the samples were filtered out, and the final matrix passed to the function. All genes from the final filtered matrix were passed to the parameter *reliable*. Further parameters were set to: *tnumber* = *Sc.tnumber*, *check* = *TRUE*, *ccl* = *150*, *mRNAs* = *60000*, *condition* = "*real_data*", *ratiomethod* = "*bias*", and time in the *phenomat* object was set to 6. Final synthesis rates in mRNA per cell per minute were obtained by dividing the synthesis rates output from the *DTA.estimate* function by 150 (length of the cell cycle in minutes). The final synthesis rates are available in Dataset EV2. Comparison of synthesis rates between samples grown for 20 vs. 60 min in galactose was performed using *DTA.dynamic.estimate* functions similarly as described above with additional columns *timeframe* and *timecourse* in the *phenomat* object specifying 20 vs. 60 min conditions. The correlation between the synthesis rates of the two time courses was calculated using the R *cor* function with *method* = "*pearson*".

## Comparison with other data

TBP residence time estimates were obtained from Zaidi et al, 2017a (Zaidi et al, 2017a), TBP and TFIIE residence time estimates from Zaidi et al, 2017b (Zaidi et al, 2017b), transcription rates from García-Martínez et al, 2004 (García-Martínez et al, 2004), and Rap1 residence times from Lickwar et al, 2012 (Lickwar et al, 2012). Correlations were calculated with R *cor* function. For residence time correlations, where we do not have exact time estimates for fast sites, Pearson's correlation was used, while for synthesis rates, Spearman's rank correlation was used.

## Model plotting

Examples of model fits were obtained by extracting Hill equation coefficients, as described in "Fitting additional reliably fast sites" section of Materials and Methods. Output model values and the measured competition ChIP (CC) values were both scaled to range between zero and one to create comparable plots by dividing the values by the estimated $X_{cc}$ parameter.

## Visual inspection with genome browser

To view the normalized HA/Myc ratios in the genome browser, BAM alignment files were first converted to bigWig files using the deepTools (3.3.1) (Ramírez et al, 2014) *bamCoverage* function. The parameter *scaleFactor* was set to per million mapped reads scaling factor for the Myc samples and to per million mapped reads multiplied by HA/Myc protein induction ratio for the HA samples. The final $\log_2$ transformed ratios of HA/Myc were obtained by passing the generated bigWig files to the deepTools *bigwigCompare* function with parameter *operation* set to *log2*.

## Residence time vs. synthesis rate

To explore the residence times of each analyzed GTF in relationship to synthesis rates, synthesis rates were first divided into quartiles using the R *ntile* function with the parameter *ngroups* set to 4. Residence times within each synthesis quartile were plotted as boxplots with ggplot2 (3.3.6) (Wickham, 2016) *geom_boxplot* function, where the middle line represents the median, the lower and upper hinge represent the first and third quartiles, and the whiskers represent 1.5 * interquartile range of the values. Normality was tested with q-q plots. Statistical testing between pairs of groups was performed using unpaired two-sided Wilcoxon tests, and an overall summary for each GTF with Kruskal–Wallis test.

The correlations between synthesis rates and residence times were calculated with the R *cor* function with *method* set to "*pearson*".

Linear models between synthesis rates were built with the R *lm* function either as linear models between synthesis rate and residence times of individual GTFs or as a linear model between synthesis rates and a linear combination of residence times of all factors in one model.

## Transcription efficiency

Transcription efficiency (TE) was obtained by multiplying the synthesis rate by the residence time of a given TF. The $\log_2$ transformed values were plotted with the ggplot2 *geom_violin* function to better represent the efficiency of a binding event to produce an RNA molecule (values below zero represent multiple binding events for RNA molecule synthesis). Medians of the log2 transformed TE values for each TF were added to the violin plots with the tidyverse (1.3.1) (Wickham et al, 2019) *stat_summary* function with parameter *fun=median*.

## PCA

To represent genes or GTFs using their corresponding high dimensional data in low-dimensional space, we performed PCA on the residence times with or without the exclusion of the reliably fast sites. Since the residence time estimates for all TFs were not available for all genes, the missing values were imputed with the missMDA (1.18) package (Josse and Husson, 2016). The table containing the reliable residence times was first passed to the *estim_ncpPCA* function with the parameter *method.cv* set to "*Kfold*". The residence timetable was then passed to the *imputePCA* function along with the *ncp* object outputted from the *estim_ncpPCA* function. The *completeObs* object from the outputted list was then passed to the *prcomp* function with parameter *scale.=TRUE* to obtain the principal components. Depending on the orientation of the input matrix passed to the *prcomp* function, principal components representing genes or GTFs were obtained. To color-code the PCA plot with mean synthesis rates, the tidyverse (1.3.1) (Wickham et al, 2019) function *stat_summary_2d* was used

with parameter $z$ set to the synthesis rates and parameter color set to "transparent". Viridis (0.6.2) (Garnier et al, 2024) color scale "B" was used for coloring. The first two principal components from the "gene-oriented" PCA matrices were then correlated with the residence times of each TF and with the synthesis rates using the R function *cor* with *method = "pearson"*.

## Residence time and synthesis rate comparison between gene classes

The list of genes with TATA-containing promoters was obtained from (Rhee and Pugh, 2012). Genes were classified as ribosomal subunits if their systematic name started with "RPL". To compare the residence times and synthesis rates between classes, an unpaired two-sided Wilcoxon test (normality tested with q-q plots) was carried out with results plotted using the ggpubr (0.4.0) (Kassambara, 2020) *stat_compare_means* function with default parameters. To compare residence times across synthesis quartiles, synthesis rates were separated into the four quartiles based on synthesis rates within each group (e.g., TATA-containing and TATA-less). Box plots were created using ggplot2 (3.3.6) (Wickham, 2016) *geom_boxplot* function, where the middle line represents the median, the lower and upper hinge represent the first and third quartiles, and the whiskers represent 1.5 * interquartile range of the values.

## Heatmap

Only genes for which residence times were available across all GTFs (except for excluded TBP, whose residence times are mostly <1 min and would therefore present mostly randomly generated values) were included in the heatmap ($n = 1417$). Reliable fast residence times were replaced by randomly generated values between zero and one (function *runif: min = 0, max = 1*; *set.-seed(42)*). Prior to plotting, residence times for each factor were z-score normalized using the R function *scale* with default settings. A final heatmap was created with the ComplexHeatmap (2.6.2) (Gu et al, 2016) function *Heatmap* with parameters set to *clustering_method_rows = "ward.D"*, *row_split = 10*. Genes belonging to each of the ten clusters (Dataset EV3) were extracted from the heatmap object, and mean synthesis rates for each cluster were calculated.

## Functional annotation

Genes belonging to each heatmap cluster were passed to g:Profiler (Raudvere et al, 2019) for pathway enrichment. In g:Profiler, *S. cerevisiae* S88C was selected as an organism, and data sources were set to GO molecular function (GO:MF), GO biological process (GO:BP), KEGG, WikiPathways (WP), and TRANSFAC. Additionally, genes from the clusters were tested for enrichment within genes associated with DNA-binding factors (DBFs) from (Rossi et al, 2021), here referred to as Yeast Epigenome database (see section "Yeast DBF database (Yeast Epigenome)" of the Materials and Methods for information about data accessions and curation). Enrichment was established by performing Fisher's exact test (R function *fisher.test*, parameter *alternative = "greater"*), where the universe was set to the union of all genes involved in the heatmap and all genes associated with a given factor. Final $p$ values were

corrected for multiple testing with false discovery rate (FDR, R function *p.adjust: method = "fdr"*). Results with FDR $p_{adj} < 0.05$ or $p < 0.05$ were considered significant.

## Yeast DBF database (Yeast epigenome)

BED files from (Rossi et al, 2021) were obtained from Gene Expression Omnibus under accession number GSE147927. Replicates were merged with the bedtools (v2.29.2) (Quinlan and Hall, 2010) *merge* function after they were sorted with the base Linux *sort* function with parameters *-k1,1 -k2,2n*. Regions were then assigned to genes analogously to the assignment of the CC regions (see "Gene assignment and filtering" section of Materials and Methods). The output consists of gene lists for individual DBFs within promoter regions.

## Additional tools used

Tidyverse (1.3.1) package (Wickham et al, 2019) was used for data processing in R, ggplot2 (3.3.6) (Wickham, 2016) was used for plotting. Illustrations were made with Biorender (https://biorender.com/). Figures were assembled with Inkscape (1.0.2, https://inkscape.org/).

## Data Availability

The datasets and computer code produced in this study are available in the following databases: • SuperSeries with all data:: Gene Expression Omnibus GSE235002. • Competition ChIP-seq data: Gene Expression Omnibus GSE235000. • Nascent RNA-seq data: Gene Expression Omnibus GSE235001. • The scripts with source data are available from https://github.com/AubleLab/PIC_competition_ChIP_scripts (release v02) and https://doi.org/10.5281/zenodo.10236107.

## Peer review information

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

## Acknowledgements

We are grateful to Patrick Grant for the discussions and critical reading of the manuscript. Funding for this work was provided by the National Institutes of Health (Grant R01 GM055763 to DTA) and the Biomedical Sciences Graduate Program, University of Virginia (Wagner fellowship to KK). Sequencing was done by the UVA Genome Analysis and Technology Core (RRID:SCR_018883).

## Author contributions

**Kristyna Kupkova**: Data curation; Software; Formal analysis; Investigation; Visualization; Methodology; Writing—original draft; Writing—review and editing. **Savera J Shetty**: Validation; Investigation; Writing—original draft; Writing—review and editing. **Elizabeth A Hoffman**: Validation; Investigation; Writing—original draft; Writing—review and editing. **Stefan Bekiranov**: Conceptualization; Software; Formal analysis; Investigation; Methodology; Writing—original draft; Writing—review and editing. **David T Auble**: Conceptualization; Resources; Data curation; Formal analysis; Supervision; Funding acquisition; Validation; Investigation; Methodology; Writing—original draft; Writing—review and editing.

## Disclosure and competing interests statement

The authors declare no competing interests.

# Expanded View Figures

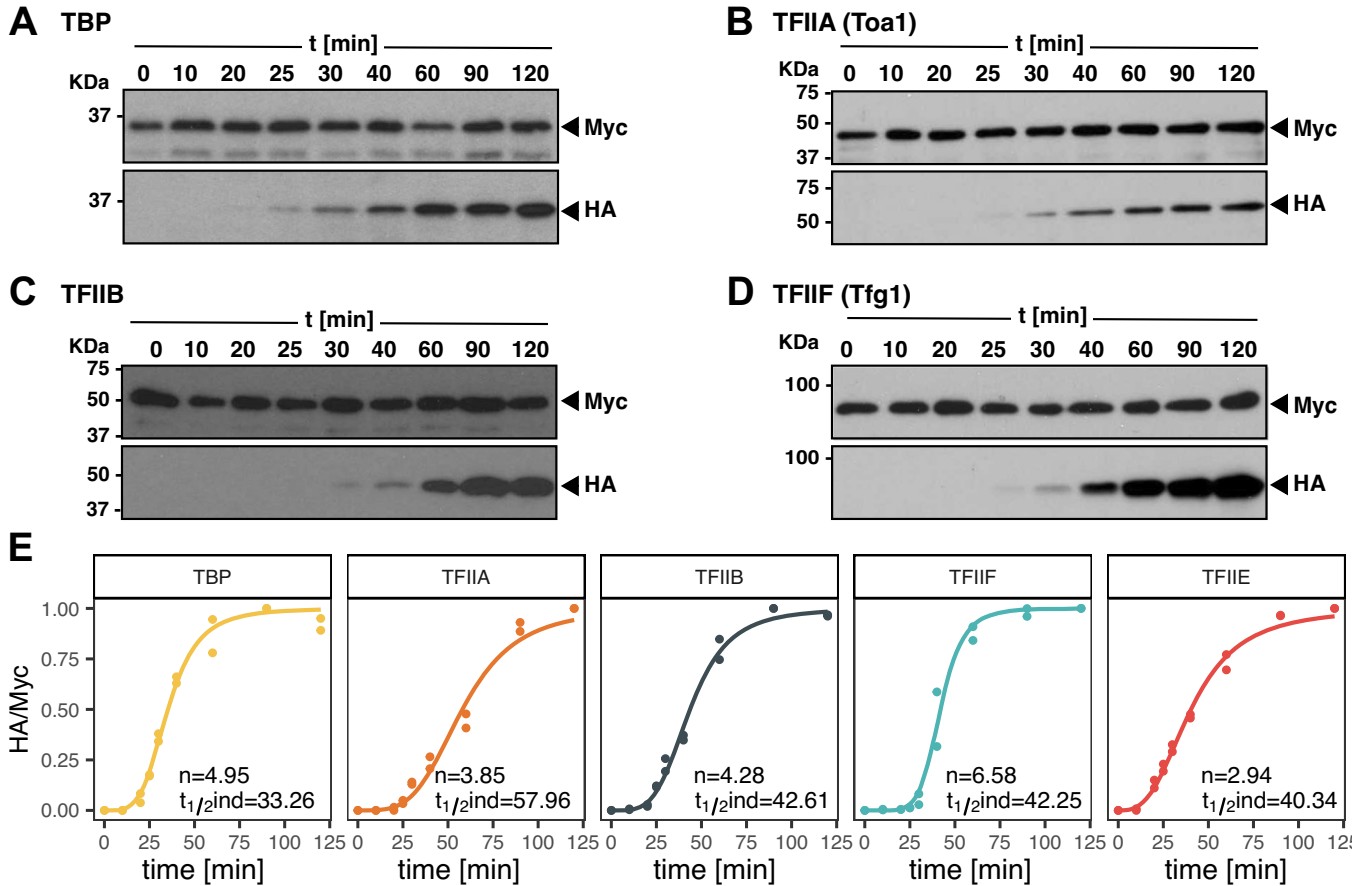

**Figure EV1.  Protein induction.**

(A–D) Western blots of (**A**) TBP, (**B**) TFIIA (Toa1), (**C**) TFIIB, and (**D**) TFIIF (Tfg1) over the indicated time course. Myc tag indicates proteins made from genes expressed under the control of their endogenous promoters, and the HA tag measures the level of the competitor expressed under galactose control. (**E**) Normalized HA/Myc ratios quantified from Western blots ($n = 2$ biological replicates) with Hill fits. Hill fit parameters are shown in the bottom right corner of each panel, n Hill coefficients, $t_{1/2}$ind half-time of HA-tagged protein induction. Source data are available online for this figure.

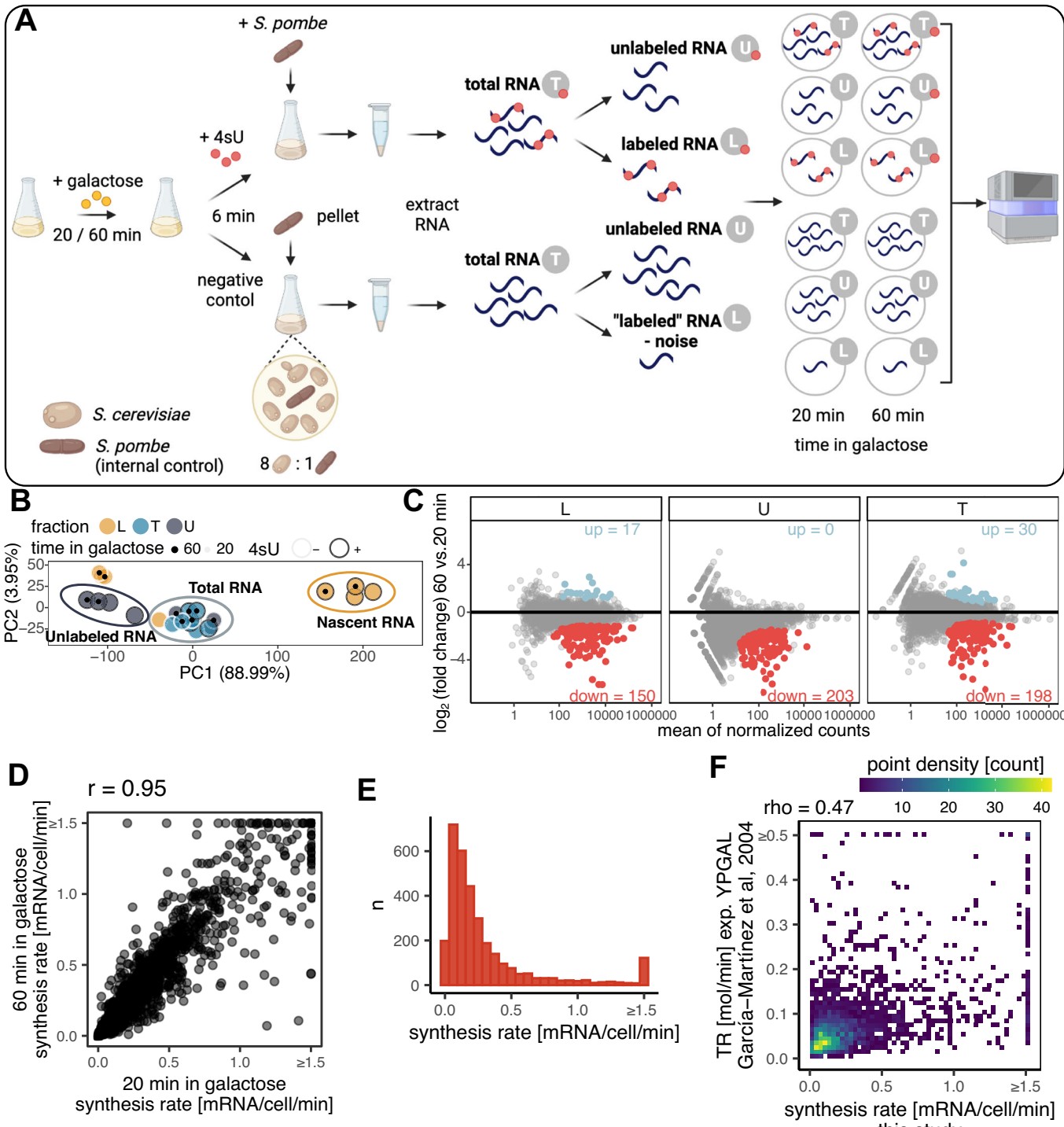

◄  **Figure EV2.  Synthesis rate estimation with dynamic transcriptome analysis (DTA).**

(**A**) Schematic overview of the DTA method as adapted in this study. (**B**) Principal component analysis (PCA) plot showing the first two principal components (PCs) calculated from normalized read coverage signal from all samples generated in this study. Highlighted are clusters of samples representing nascent RNA (L fraction after 4-sU addition), total RNA (T fraction after 4-sU addition as well as from negative control, along with U fraction from negative control), and unlabeled RNA (U fraction after 4-sU addition). Percentages within the axis labels indicate the percentage of variance explained by a given PC. (**C**) MA plot showing differentially expressed genes between samples grown for 60 vs. 20 min in galactose. Each point represents a gene, the x-axis indicates the size of a given gene in terms of the mean number of reads after normalization mapped to the gene, and the y-axis shows $\log_2$ of fold change between the two conditions. Highlighted are significantly misregulated genes, blue: upregulated at 60 min, red: downregulated at 60 min compared to the 20-min time point, significance threshold: FDR-corrected $p$ value ($p$adj) <0.05. (**D**) Comparison of synthesis rates (in mRNA per cell per minute) estimated from samples grown for 20 min in galactose (x-axis) vs. 60 min in galactose (y-axis). Pearson's correlation coefficient can be found above the plot. (**E**) Histogram showing the distribution of synthesis rates (in mRNA per cell per minute) estimated jointly from samples grown for 20 and 60 min in galactose. Synthesis rates higher than 1.5 were combined into one bar to eliminate long tails. (**F**) Comparison of synthesis rates generated in this study (x-axis) to those generated by García-Martínez et al, 2004 (García-Martínez et al, 2004). Spearman's correlation coefficient is mentioned above in the plot. The plot is color-coded based on point density in each area. Symbol ≥ on the axis indicates that values higher than an indicated value were shrunk for plotting purposes to eliminate outliers.

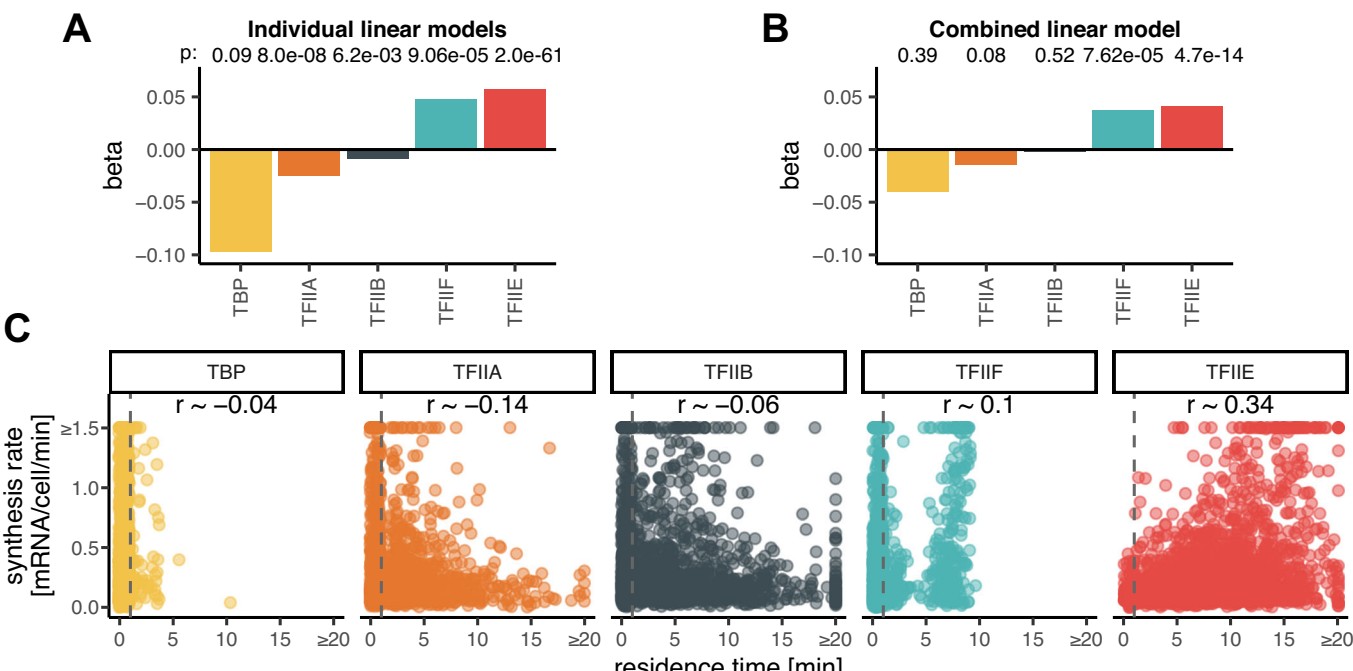

**Figure EV3.  Comparison between residence times and synthesis rates.**

(A) Bar plot showing β (beta) coefficients for linear models built between synthesis rates and residence times of an indicated GTF (synthesis rate ~ $\beta$*res. time$_{GTF}$ + α). (B) Analogous to (A), showing coefficient from a linear model combining all factors (synthesis rate ~ $\beta_{TBP}$*res. time$_{TBP}$ + $\beta_{TFIIA}$*res. Time$_{TFIIA}$ + $\beta_{TFIIB}$*res. Time$_{TFIIB}$ + $\beta_{TFIIF}$*res. Time$_{TFIIF}$ + $\beta_{TFIIE}$*res. Time$_{TFIIE}$ + α). (C) Relationship between residence times (x-axis) and synthesis rates (y-axis) for GTFs as indicated. Pearson's correlation coefficient estimates, r, are indicated in each panel. Symbol ≥ on the axis indicates that values higher than an indicated value were shrunk for plotting purposes to eliminate outliers. In the plots, gray dashed line separates values randomly generated in this study for reliably fast sites.

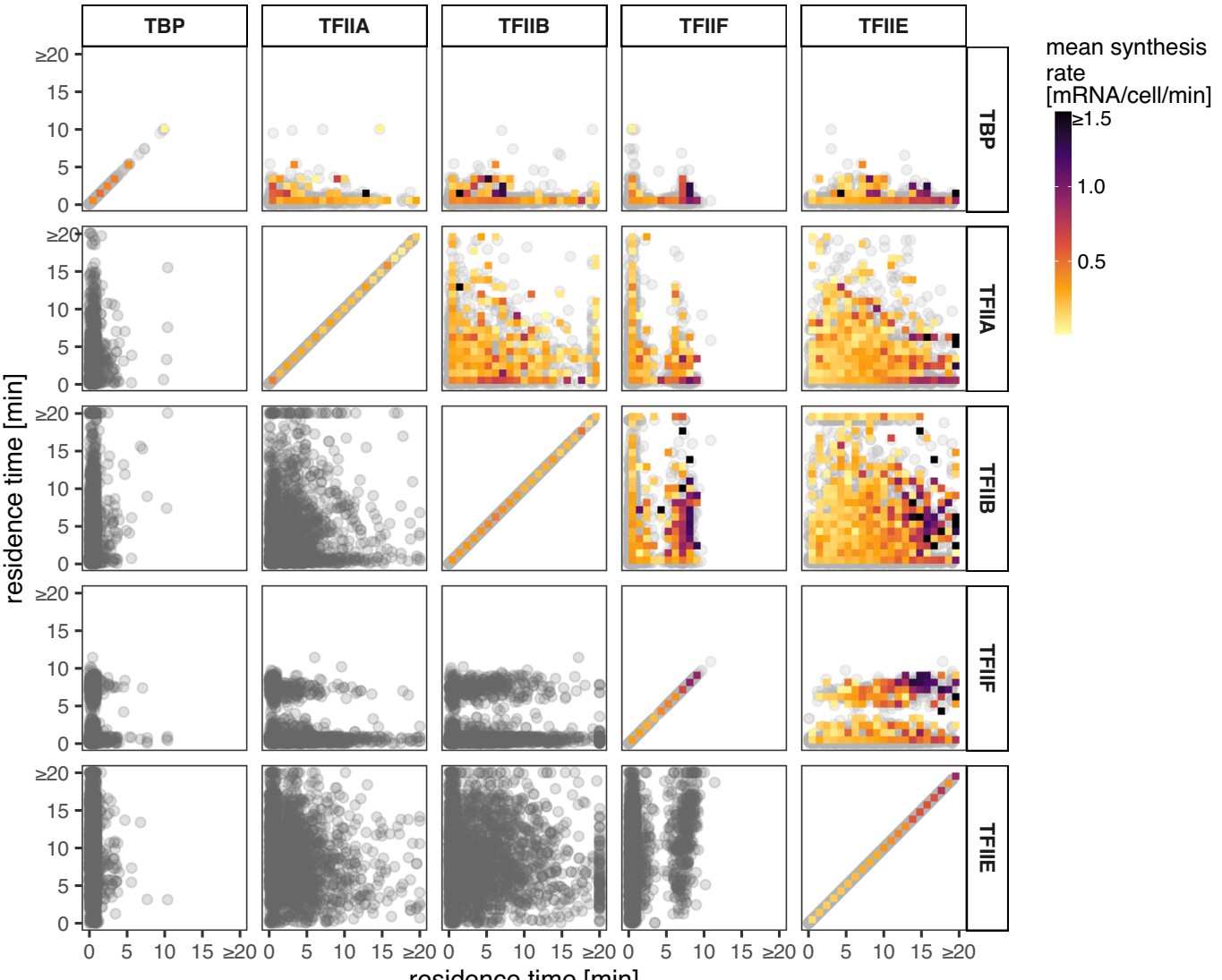

**Figure EV4. Relationships among GTF residence times and to mRNA synthesis rates.**

Each panel shows a comparison of residence times of pairs of GTFs as indicated in the panel titles. Each point is a shared gene target. The color map shows the mean synthesis rates of the genes under the given area. Source data are available online for this figure.

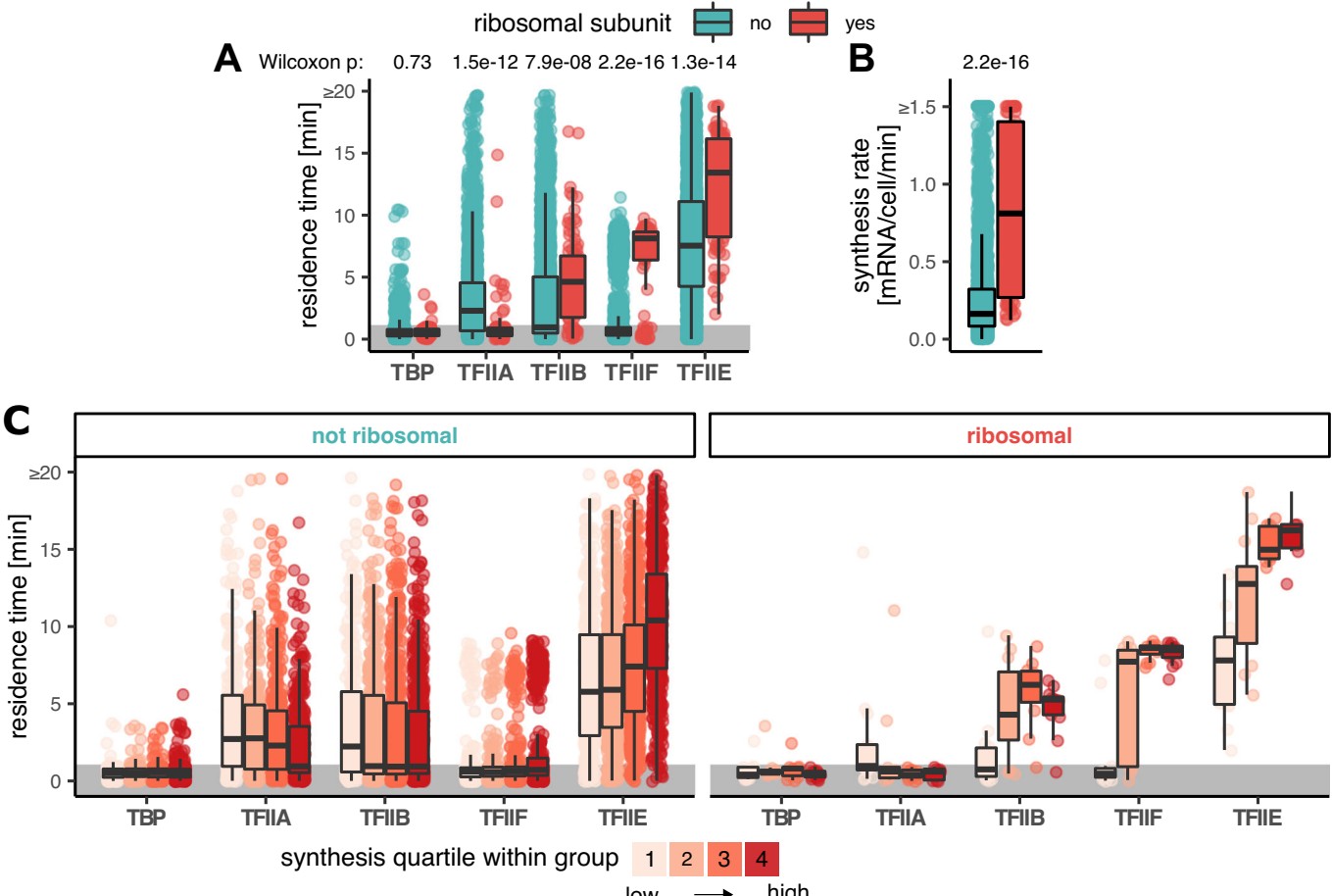

**Figure EV5. Comparison of residence times between genes coding for ribosomal subunits and others.**

(A) Box plots showing the comparison of residence times (y-axis) for a given GTF (x-axis) for genes coding for ribosomal subunits (red) and other genes (green). The number of observations (*n*): not ribosomal/ribosomal TBP- 2810/87; TFIIA- 2418/76; TFIIB- 3420/93; TFIIF- 2776/88; TFIIE- 3723/86. (B) Box plot showing the comparison of synthesis rates (y-axis) between genes coding for ribosomal subunits and other genes. Wilcoxon *p* value is indicated. Number of observations (*n*): not ribosomal- 3169, ribosomal- 57. (C) Box plots showing the comparison of residence times (y-axis) for a given GTF (x-axis) across synthesis quartiles within genes coding for ribosomal subunits and other genes. In the plots, the gray area highlights values randomly generated in this study for reliably fast sites. Symbol ≥ on the axis indicates that values higher than an indicated value were shrunk for plotting purposes to eliminate outliers. The number of observations (*n*): not ribosomal/ribosomal TBP- Q1: 209/11, Q2: 320/10, Q3: 414/11, Q4: 583/14; TFIIA- Q1: 231/13, Q2: 316/9, Q3: 382/13, Q4: 409/9; TFIIB Q1: 293/14, Q2: 475/12, Q3: 569/11, Q4: 589/12; TFIIF- Q1: 245/11, Q2: 355/12, Q3: 419/12, Q4: 494/12; TFIIE- Q1: 403/13, Q2: 577/12, Q3: 598/10, Q4:540/10. Data information: (A–C) In boxplots, the middle line represents the median, the lower and upper hinges represent the first and third quartiles, and the whiskers represent the 1.5 * interquartile range.

