## [Peer Review File · The EMBO Journal]

Genome-scale chromatin binding dynamics of RNA Polymerase II general transcription machinery components

Kristyna Kupkova, Savera Shetty, Elizabeth Hoffman, Stefan Bekiranov, and David Auble

Corresponding author(s): David Auble (auble@virginia.edu)

Review Timeline:

Submission Date:	2nd Aug 23
Editorial Decision:	22nd Sep 23
Revision Received:	14th Dec 23
Editorial Decision:	12th Feb 24
Revision Received:	20th Feb 24
Accepted:	28th Feb 24

Editor: Cornelius Schneider

Transaction Report:

Dear Dr. Auble,

Thank you for submitting your manuscript for consideration by the EMBO Journal. It has now been seen by three referees whose comments are shown below.

Given the referees' positive recommendations, I would like to invite you to submit a revised version of the manuscript, addressing the comments of all three reviewers. I should add that it is EMBO Journal policy to allow only a single round of revision, and acceptance of your manuscript will therefore depend on the completeness of your responses in this revised version. Given that referees #1 and #3 felt that the manuscript would benefit from the addition of other factors, namely TFIID/TAFs and TFIIH subunits (referee #1) and TAFs, IIF, Mediator or Pol II (referee #3). Given that I had similar thoughts during my initial assessment of the manuscript I have also asked all referees to comment on the requirement for inclusion of additional transcription factors into the analysis. You can find this additional discussion below for your information. In light of these considerations I think it would be helpful to per-discuss the revision plan by videoconferencing or email. Please let me know whichever you prefer.

Thank you for the opportunity to consider your work for publication. I look forward to your revision.

Yours sincerely,

Cornelius Schneider, PhD
Editor
The EMBO Journal
c.schneider@embojournal.org

We realize that it is difficult to revise to a specific deadline. In the interest of protecting the conceptual advance provided by the work, we recommend a revision within 3 months (21st Dec 2023). Please discuss the revision progress ahead of this time with the editor if you require more time to complete the revisions. Use the link below to submit your revision:

Referee #1:

In this manuscript Kupkova et al. estimate site-specific, genome-scale chromatin binding dynamics or residency times of five general transcription factors (GTFs): TBP, TFIIA, TFIIB, TFIIE and TFIIIF in budding yeast by using competition chromatin immunoprecipitation (CC). In addition, the authors compared promoter binding dynamics of the studied GTFs with RNA synthesis rates to determine how the chromatin binding of these GTFs relates to the production of RNA. From these and additional analyses the authors conclude that they provide a "rich resource for exploring the mechanistic relationships between PIC assembly, gene regulation, and transcription.

Major Concerns:

- The CC method used by the authors is not supposed to provide information about pre-initiation complex (PIC) assembly. PICs are formed only when genes are turned on at the first time, but as the authors monitor GTF exchange (old versus newly synthesized) at expressed genes they are only obtaining information about re-initiation complexes. Thus, throughout the manuscript the authors should avoid the use of PIC assembly and other related expressions.
- As the authors test TFIIA, TFIIE and TFIIIF, GTFs composed by several subunits, they should also test TFIID/TAFs and TFIIF subunits.
- Figure 2A and Fig S1a-d, why are the induction times of the different GTFs quite different? TFIIE starts at 20 min, while TFIIA starts to express at about 40 min. Also the newly synthesized TFIIA (which subunit?) is very weak compared to the others. Do these differences influence the CC measurements?
- The western blot figures should contain molecular markers.
- All the western blots panels should be presented with a loading control.
- To provide a more accurate estimate for the base of competition in the ChIP experiments the authors should present western blots on nuclear extracts.
- Along the same lines, have the authors considered/controlled the maturation/nuclear import time for the GTFs analyzed? The kinetic analysis uses the protein induction (accumulation) curve time-course as a baseline to define immediate vs delayed chromatin binding. Would the analysis be biased if, for instance, a factor like TFIIE would require more time to become functional upon completion of protein synthesis? For instance, to complete folding, nuclear import, post-translational modifications or integration with other subunits?
- What would be the molecular interpretation of TFIIE having a much higher residence time when compared with the other tested GTFs?
- The authors should comment on the apparent difference in residence times between the present study and that of the Carl Wu's study (Nguyen et al., Molecular Cell, 2021) which employed live-cell imaging approaches (single-particle tracking) on the same factors than investigated here. Specifically, the latter study finds very long-lived binding events for TBP (unresolved from H2B control) and rather short-lived binding events for TFIIE (~2 s). Apparently, these observations look opposite to the findings of the present study, where TBP and TFIIE show fast and slow turnover dynamics, respectively.
- In Fig. 5a: how do the authors explain cases where "late" GTFs, such as TFIIIF and TFIIE, show very stable chromatin binding while, on the same sites, "early" GTFs (TBP, TFIIA, TFIIB) show very short-lived binding? For example, in cluster 1 how would TFIIIF and TFIIE have a slow exchange without TBP, TFIIA and TFIIB?
- The model presented in Fig. 6 does not seem to summarize the obtained data, and does not help in the understanding of the new concept(s). The authors should either remove it or present it in a more comprehensive way.
- The authors' model works only if the CC method detects a real exchange between the "old" and the "new" factor. What if the induced protein signal increases, even when the residing protein signal does not decrease and remains constant? Thus, it would not be an exchange nor a competition. Can the authors show that the "old" factor ChIP signal decreases, when the newly synthesized factor arrives?

Referee #2:

Kupkova et al. use competition ChIP assays in yeast to measure residence times of the Pol II general transcription factors (TBP, TFIIA, TFIIB, TFIIF, and TFIIE) in vivo on a genome-wide scale. The approach involves inducing an HA-tagged form of the relevant GTF using a GAL promoter in a diploid strain that constitutively expresses a Myc-tagged version of the same GTF from the other allele. By calculating HA/Myc ChIP signals over induction time, the authors compute residence times for each of the GTFs at ~3000 promoters. Elevating the impact of the work, the authors performed dynamic transcriptome analysis to determine RNA synthesis rates and then compared these rates to the factor residence times, leading to measurements of transcription efficiency (i.e. mRNA molecules per GTF binding event). Primary conclusions are (1) residence times of most GTFs examined, notably TBP, TFIIA, and TFIIB, at many promoters are short in vivo (<1 min) and do not determine transcription rates; (2) residence times for TFIIE and a subclass of TFIIF binding events are considerably longer; (3) genes with the longest TFIIE residence times are associated with the highest transcript synthesis rates; (4) transcription rate is positively correlated with longer residence time for TFIIE and/or TFIIF; (5) enrichment analysis showed, among other relationships, that ribosomal protein genes are enriched among the class of genes with the longest TFIIE/TFIIF residence times and highest mRNA synthesis rates.

The paper describes a novel approach toward measuring Pol II PIC assembly kinetics in vivo at thousands of promoters. The approach provides a valuable alternative to other methods for monitoring transcription factor dynamics, including single molecule tracking within cells and single-molecule in vitro methods. As noted above, an especially important aspect of the work is the comparison between residence times and transcription efficiency on an individual gene level at a genome-wide scale. Overall, this is a high-quality paper that makes a novel contribution toward understanding Pol II initiation in vivo. It will be of interest to many in the Pol II transcription field.

Specific comments:

1. The Buratowski lab (Baek et al. 2021, Mol Cell) has used single-molecule TIRF microscopy to monitor the kinetics of GTF binding to a model gene in vitro and observed frequent dissociation of TFIIE over the time course of their experiments. They concluded that TFIIE association was unexpectedly dynamic. The authors have not cited this paper or provided an explanation for the apparent discrepancy.
2. One other surprising aspect of the Baek et al. paper was the finding of GTFs at UAS elements. Are the authors' ChIP data of sufficient resolution to measure residence times at UAS elements?
3. Because the ChIP and DTA experiments involve galactose induction, there is at least one group of genes undergoing temporal activation (the GAL genes). Can the authors comment on the residence times of the GTFs at these genes and whether they differ from those for constitutively expressed genes in their data set?
4. The use of the term Hill equation and Hill coefficient is confusing, as the Hill coefficient is most often used to describe binding cooperativity.
5. By what measure did the authors confirm that the tags on the GTFs did not affect activities of the factors? Data showing the tags are functionally inert are needed.
6. Figure 2E and S2B--- Are these data from one replicate or an average of more than one replicate?
7. Figure 2F--- Define "density" in the legend.
8. Figure 3C--- This is a key result. Measures of statistical significance are needed.
9. Figure 4 is introduced as a pair-wise comparison of the GTF residence times; however little discussion is given to these comparisons. What is the value of these GTF-GTF comparisons? Panels in S5 might be more relevant to include in the main body.
10. Figure S1--- Any thoughts on why TFIIA and TFIIE levels accrue more slowly upon GAL induction than the other factors? Also, panel E in this figure requires a label on the x-axis.
11. Figure 3 panels E-G vs. Figure S4 A-C. The distinctions between these figures should be made clearer to the reader.

Referee #3:

Auble's group used competition ChIP to determine the dynamics of TBP, TFIIA, TFIIB, TFIIF and TFIIE binding to pol II promoters in yeast. Overall, the study is rigorous, novel and the lab has experience with the methodology and interpretation. The findings are consistent with TFIIE and F dynamics playing more notable roles in PIC function genomewide highlighted by an important correlative role with nascent transcription. I was a little disappointed at the narrowness of the study - no TAFs, IIH, Mediator or Pol II-- but the primary and supplemental figures displayed a huge amount of data representing even huger amounts of good work.

I thought the model the group used to handle the synthesis of the competitor factors versus displacement was appropriate.

I noted that the authors expressed the other subunits of IIA, IIF and IIE simultaneously although I question why they felt that necessary. Clearly some IP and blots could have determined whether the HA tagged subunit readily exchanged into the dimeric GTFs. Additionally, I had trouble understanding how the final GAL-stimulated levels of HA-GTF matched the total level of the HA+Myc GTF in cells, e.g., normal TBP level versus induced level as measured with a TBP antibody. There was a normalization

process used in the displacement assays but the level of overexpression of one GTF versus the other might influence the dynamics and although I am somewhat satisfied with the authors treatment of this issue, I was wondering whether the rightward shift in the curves could be affected or correlated with higher concentrations of the final TFIIF and TFIIE levels versus TFIIB and TBP, which are single subunits.

Would have liked to see some specific browser tracks displaying the dynamics for all the factors at the same promoter analogous to what was shown in Supplemental 1 but much more thorough. This would have been particularly interesting on the Ribosomal protein genes, where Tony Weil earlier showed an effect of Rap1 on the TFIIA-TFIID complex and argued for interactions between Rap1, TFIIA and TAF4 (PMC3743499). This is particularly interesting because others have shown in the human PIC that TFIIA interacts with TAF4 within the high res 3D structure. I have a hunch the TAFs might be a better indicator of TFIID occupancy.

Otherwise this is a good, solid paper.

Additional discussion between the referees regarding the requirement for additional transcription factors:

Referee #1

The authors claimed that they did not test multisubunit complex (such as TFIID, TFIIH, Mediator and/or Pol II) subunits because they could not test/control the incorporation of the newly synthesized subunit(s) in these large multiprotein complexes. Nevertheless, they have tested and analysed yeast TFIIA, TFIIF and TFIIE, which are composed by two subunits each, opening the question what is the difference between a newly synthesized factor incorporating in a two subunit GTF, or incorporating in a larger multisubunit GTF?

I'm almost sure that the authors have the yeast strains for doing these experiments, or even they have even done the corresponding experiments, but for some reasons they did not want to show them. They could at least test one TAF and one of the TFIIH CAK (three subunit complex) subunits.

Otherwise, we could accept a reasonable clear and logical explanation about why they did not do these experiments.

Referee #2

For TFIIA, TFIIF, and TFIIE, which are hetero-dimeric, the authors expressed both subunits under GAL control with only one subunit being HA-tagged. The rationale was to balance expression of the two subunits, since GAL induction leads to overexpression. However, they didn't show western data confirming equal levels of subunit induction. I suspect the authors did not analyze larger complexes to avoid putting all subunits under GAL control. This would be possible but labor-intensive and it's unclear if the subunits would indeed be balanced in the end. This is my best guess. The authors should explain more clearly why they did not test association of TFIIH and other relevant complexes. With respect to the roles of coactivators and Rap1 in controlling GTF residence times, this is one of the next big questions to test with this system and they will need to generate some mutants. I don't think this will be necessary for the current paper.

Referee #3

There is a lot of work here. I agree that it would be ideal to see TFIID, H and Mediator as part of this study, but it would be asking a lot due to the number of subunits. They forced themselves into a corner by designing the study to include overexpression of the partner subunits. I would like an explanation for why they did that and was it necessary. Minimally, I agree that blots of the native TFs would be helpful to assess overall levels. But my main concern is whether this overexpression affects the kinetics. Overall though, I think this paper is a good start.

This letter accompanies the resubmission of our manuscript entitled, "Genome-scale chromatin binding dynamics of the RNA Pol II general transcription machinery components" by Kupkova et al, which we ask to be considered for publication in the Journal. This version of the manuscript has been reformatted to align with EMBO Journal formatting requirements.

First, we express our gratitude to the reviewers for their positive reviews and for their suggestions to improve the study. Notably, Reviewer 2 stated, "Overall this is a high-quality paper that makes a novel contribution toward understanding Pol II initiation *in vivo*. It will be of interest to many in the Pol II transcription field." Reviewer 3 commented, "Overall, the study is rigorous, novel and the lab has experience with the methodology and interpretation.The primary and supplemental figures displayed a huge amount of data representing even huger amounts of good work. A good solid paper." The reviews are very thoughtful, and indeed, reviewers raised a couple points that we hadn't thought of. The reviewers appreciated the scope and scale of the work that this paper represents- this paper does represent a tremendous amount of work. Including all the work that was done to establish and validate the system, it represents nearly the entire output of my lab during the prior funding period of my longstanding NIH grant. In addressing the reviewers' comments, we feel the revised manuscript is notably stronger.

Below I detail specific responses to each comment by the reviewers.

Yours sincerely,

David T. Auble

Reviewer1

Major Concerns:

1. The CC method used by the authors is not supposed to provide information about pre-initiation complex (PIC) assembly. PICs are formed only when genes are turned on at the first time, but as the authors monitor GTF exchange (old versus newly synthesized) at expressed genes they are only obtaining information about re-initiation complexes. Thus, throughout the manuscript the authors should avoid the use of PIC assembly and other related expressions.

By using the term "preinitiation complex" we are referring to the complex that forms on the promoter prior to the synthesis of a molecule of RNA. It is certainly possible that there are different factors at different promoters that contribute to reinitiation versus the synthesis of the initial RNA that is synthesized when a promoter is first activated, but it is clear that all five of the factors that we have investigated are essential for all Pol II transcription in yeast (Petrenko et al, eLife 8:e43654, 2019). For this reason, we posit that all of these factors are involved in all reinitiation as well as the first round of RNA synthesis (at promoters that are regulatable) *in vivo*, and that their requirement is indicative of the formation of a complete complex, a PIC, that is structurally and biochemically similar and that precedes the initiation of

synthesis of RNA. In the revision we have clarified what we mean by the term PIC as follows (lines 41-44):

As these factors participate in all Pol II-mediated transcription in yeast, we use the term 'PIC' in this paper to refer to complexes that catalyze the initiation of transcription including during the first round of synthesis upon promoter activation as well as reinitiation or ongoing initiation at unregulated promoters.

2. As the authors test TFIIA, TFIIIE and TFIIF, GTFs composed by several subunits, they should also test TFIID/TAFs and TFIIH subunits.

The analysis of TFIIA, TFIIIE, and TFIIF was made possible in our view by the ability to express both subunits of these stable biochemically-defined entities. From the comments of the Reviewer, we realize that we did not make this clear, and in the revision we have expanded the text to clarify this. (See lines 331-337):

Given the complexity in interpreting the results from a CC experiment in which one overexpressed a single or a few subunits of a multi-subunit complex, it would be difficult if not impossible to apply this method to the analysis of multisubunit complexes as currently implemented. In future work and using methods suitable for analysis of multi-subunit complexes, it will be interesting to investigate the dynamics of TFIID, TFIIH (Greber et al, 2019; Nogales & Greber, 2019), Mediator and Pol II itself (Nozawa et al, 2017; Plaschka et al, 2015).

In support of this, Reviewer 2 states our rationale well in the follow-up discussion among the reviewers about the need for additional data, indicating that, "For TFIIA, TFIIF and TFIIIE, which are hetero-dimeric, the authors expressed both subunits under GAL control with only one subunit being HA-tagged.... I suspect the authors did not analyze larger complexes to avoid putting all subunits under GAL control. This would be possible but labor-intensive and it's unclear if the subunits would indeed be balanced in the end. This is my best guess. The authors should explain more clearly why this did not test association of TFIIH with the other relevant complexes. With respect to the roles of coactivators and Rap1 in controlling GTF residence times, this is one of the next big questions to test with this system and they will need to generate some mutants. I don't think this will be necessary for the current paper." (We feel that the added text addresses the comment by Reviewer 2 regarding TFIIH.) Moreover, Reviewer 3 wrote in follow-up discussion that analysis of other complexes "would be asking a lot due to the number of subunits".

Reviewer 3 commented that we "forced [our]selves into a corner" by this study design, but we feel that our study design is a strength. If we had simply overexpressed one subunit of a multisubunit complex as a competitor, we (and I feel certain interested others as well) would have questioned whether the dynamics we measured were driven by a kinetic bottleneck in the exchange of the competitor subunit with existing complexes or reflected true exchange as we have concluded. We are very interested in the chromatin binding kinetics of TFIID, TFIIH, etc.,

but we feel that our analysis of these five key factors (the minimal set required for in vitro transcription as defined 30 years ago (Tyree et al, G & D 7:1254, 1993), plus TFIIA) constitute a complete story. In line with some of the reviewer comments, we feel that the analysis of additional complexes is appropriately the work of the next study for which our paper establishes a very solid foundation.

3. Figure 2A and Fig S1a-d, why are the induction times of the different GTFs quite different? TFIIIE starts at 20 min, while TFIIA starts to express at about 40 min. Also the newly synthesized TFIIA (which subunit?) is very weak compared to the others. Do these differences influence the CC measurements?

The short answer is that we don't know why the induction times vary. We presume that this is indicative of differences in rates of synthesis post production of the RNA since the promoters are the same. The differences do not influence the CC measurements; we use the induction time course for each competitor GTF to estimate residence times. What matters in our approach is whether the kinetics of exchange (i.e., the normalized CC ratio data) are significantly different from the kinetics of competitor synthesis (i.e., the normalized protein ratios). This is the main insight that allows us to measure residence times as short as a minute or two even though the production of competitor takes much longer.

First, we clarified details associated with the strain construction to address the question of which subunits were tagged (lines 364-371):

The parental diploid strain W303 (Ralser et al, 2012) was used to generate all of the competition CHIP strains. For each GTF, one allele was N-terminally tagged with 3xHA and placed under the control of an inducible GAL1 promoter. The other allele was N-terminally tagged with 9xMyc and remained under the control of the endogenous promoter (Longtine et al, 1998). For measurement of GTFs with two subunits, one allele of each subunit was placed under GAL1 control and one subunit was tagged with Myc or HA. For TFIIA, the Toa1 subunit was epitope-tagged; for TFIIIE, the Tfa1 subunit was tagged, and for TFIIIF, the Tfg2 subunit was tagged.

We also clarified which subunit was tagged for TFIIA, TFIIIE, and TFIIIF in the legend to Fig 2C. In addition, this text has been added (lines 408-415):

Although the galactose-induction experiments were conducted in the same way for each strain, for unknown reasons we observed reproducible differences in the time course of accumulation of the competitor (Fig 2C). Since the promoter is the same for each factor, we presume this reflects differences in the rate of protein synthesis in vivo post-transcription. The CC method relies on measuring a difference between the rate of HA/Myc isoform turnover versus the rate of synthesis of the competitor (HA isoform), and for this reason, factor-specific differences in the rate of competitor synthesis do not impact the results reported here.

4. The western blot figures should contain molecular markers.

These have been added to the figures in this revision.

5. All the western blots panels should be presented with a loading control.

The native promoters of the GTFs examined in this study are not controlled by galactose, so we consider the Myc-tagged alleles to effectively be the loading controls. However, we also assessed (and confirmed) equivalent protein loading across samples by staining of the blots with Ponceau S. In this submission, we provide a few images of the Ponceau S-stained blots for the reviewer to consider. We did not feel that they add enough to the manuscript to warrant including them in the revision, although we would be more than happy to add them to the supplemental material if the reviewer feels that is important.

6. To provide a more accurate estimate for the base of competition in the ChIP experiments the authors should present western blots on nuclear extracts.

We understand what the reviewer is driving at; this is an interesting idea. We don't feel that such experiments would be informative, however, because it takes time to prepare nuclear extracts- much longer than the difference in time between individual time points. As a result, the temporal resolution would be lost during nuclei isolation from cells that are otherwise still alive and metabolically active.

7. Along the same lines, have the authors considered/controlled the maturation/nuclear import time for the GTFs analyzed? The kinetic analysis uses the protein induction (accumulation) curve time-course as a baseline to define immediate vs delayed chromatin binding. Would the analysis be biased if, for instance, a factor like TFIIE would require more time to become functional upon completion of protein synthesis? For instance, to complete folding, nuclear import, post-translational modifications or integration with other subunits?

Based on published data (Timney et al, J Cell Biol. 175(4):579-593, 2006), nuclear import in yeast can occur rapidly, but the kinetics depend on the cargo and its concentration. It is quite possible that it takes ~minutes to import the newly made competitor into the nucleus. We don't have any data about the time it may take for assembly and maturation of the newly made competitor GTFs. These are things that we have not taken into account explicitly, but such effects are accounted for by our analytical approach. Specifically, we normalize the total read count-normalized GAL-induced HA-GTF CC data to the ratio of the GAL-induced HA-GTF protein over Myc-GTF protein levels. This effectively removes the delay and treats the cytoplasmic protein induction as though it were nuclear induction. If this normalization had not removed the delay of functional GTF in the nucleus, then our GTF protein curve should actually be shifted further to the right in time. However, we currently observe a majority of fast sites whose normalized CC curves rise with and closely match the protein induction curve. Thus, shifting the protein curve further to the right would result in the majority of sites' CC curves rising faster than the protein induction curve, which is physically impossible. Importantly, we perform the same normalization for all GTFs, yet we observe distinct dynamics for each of the factors. In order to clarify this important point, we added text and modified the text as follows (lines 494-499):

Since most GTF binding events display fast, second-timescale dynamics (Nguyen et al, 2021), this normalization method tends to bring the kinetics of fast sites in line with the time-scale of competitor induction regardless of whether there is a delay in e.g. the time it takes for competitor protein maturation or nuclear import. Notably, the turnover model assumes the protein induction is effectively occurring in the nucleus, which this normalization approximates.

8. What would be the molecular interpretation of TFIIIE having a much higher residence time when compared with the other tested GTFs?

We realized from the reviewer's comments that we did not do a sufficient job of explaining the results, our interpretation, and our model. This has prompted us to revise the model (Fig 5 – originally Fig 6) and associated text. The key point is that assembly of the transcription complex at the promoter is a very inefficient process. That being the case, the great majority of complexes formed with GTFs that bind early in the process are unstable and decay. The residence times that we measure are the overall residence times for all the complexes that are formed and contain the GTF of interest. Since most of the TBP-containing complexes (for example) do not go on to form a complex capable of producing an mRNA, the residence times are short. In contrast, TFIIIE-containing complexes are longer lived and associated with the rates of RNA synthesis because TFIIIE is predominantly found in complexes that are more stable (relatively speaking) and do go on to produce mRNA. Please see below for additional discussion and details about the changes that were made to the manuscript.

9. The authors should comment on the apparent difference in residence times between the present study and that of the Carl Wu's study (Nguyen et al., Molecular Cell, 2021) which employed live-cell imaging approaches (single-particle tracking) on the same factors than investigated here. Specifically, the latter study finds very long-lived binding events for TBP

(unresolved from H2B control) and rather short-lived binding events for TFIIE (~2 s). Apparently, these observations look opposite to the findings of the present study, where TBP and TFIIE show fast and slow turnover dynamics, respectively.

We appreciate the comment, and it points out the need for us to do a better job of integrating previous results with the results in this study. We can't comment too specifically about the results in the published paper, but we note that in most cases it can't be determined by SMT where in the genome binding is occurring or how such binding relates to transcription of particular genes, whereas we have captured binding events at promoters with measured transcriptional activity. For this reason, we argue that the binding dynamics that we report are functionally relevant. In terms of the long-lived TBP reported by Nguyen *et al*, 2021, the rapid dynamics that we report for TBP are consistent with our previously published results using FRAP. In our prior study, 90% of the photobleached TBP signal in yeast nuclei recovered in less than 5 sec and the signal was fully recovered in 15 sec or less (Sprouse *et al*, PNAS 105:13305, 2008). A key observation supporting the rapid TBP dynamics that we reported was that it was dependent on Mot1. Regarding TFIIE, it is worth noting that with competition ChIP we can only measure binding to sites that are crosslinkable. It is quite possible- perhaps likely- that binding events that are too rapid would not be efficiently measured by this method because they were not captured efficiently. (This is a point that we addressed in the original submission.) This could be one reason why we were able to identify relatively fewer sites with binding dynamics for TBP. To address this, we have added the following to the text (and please see the response to Reviewer 2, point (1) for discussion of the new text that immediately precedes this (lines 266-273):

These considerations, in addition to the focus in the present study on promoter regions, could explain, at least in part, why TFIIE was observed by single-molecule tracking to be engaged primarily in short-lived binding events whereas long-lived TBP binding events were observed (Nguyen et al, 2021). In this regard, it is worth noting that the highly dynamic behavior of TBP reported here is consistent with highly mobile TBP in the nucleoplasm overall, which was dependent on the TBP-DNA dissociating enzyme Mot1 and observed in live cells by fluorescence recovery after photobleaching (FRAP, ref. Sprouse et al, 2008).

10. In Fig. 5a: how do the authors explain cases where "late" GTFs, such as TFIIF and TFIIE, show very stable chromatin binding while, on the same sites, "early" GTFs (TBP, TFIIA, TFIIB) show very short-lived binding? For example, in cluster 1 how would TFIIF and TFIIE have a slow exchange without TBP, TFIIA and TFIIB?

This is related to the reviewer's comment (8) above regarding the sites with long-lived TFIIE and our interpretation of that observation. As described above, we interpret the differences as being due to the inefficiency of transcription complex formation and the fact that the residence times that we report are an average of the dynamics of all events that contain these GTFs. While we can't rule out the possibility of complexes that have never been observed, the structural and biochemical evidence to date support the conclusion that a promoter-bound

complex containing TFIIE will contain the other GTFs as well. We do not observe longer residence times for early-binding GTFs because such complexes only rarely lead to a functional transcription complex. We have address this in the revised text as follows (lines 256-266):

Since the formation of a PIC is mutually dependent on all of the GTFs (Petrenko et al, 2019) and structural data are consistent with the requirement for e.g. TBP and TFIIB binding to establish a platform for the binding of Pol II, TFIIF and TFIIE (Osman & Cramer, 2020; Nogales et al, 2017), it may appear counterintuitive that such “early” binding GTFs have shorter residence times than the “late” binding GTFs at many promoters. It is important to recognize that the residence times that we report here reflect the global average of the residence times for all complexes formed in vivo that contain the GTF of interest (Fig 5A,B). Thus, core promoter-bound complexes that contain TFIIE or TFIIF are highly likely to contain TBP and TFIIB. We infer that the reason there is shorter-lived TBP and TFIIB at many promoters with longer-lived TFIIF and TFIIE is that most TBP- and TFIIB-containing complexes do not lead to the formation of a complex of a productive PIC that contains TFIIF or TFIIE (Fig5C).

We also address this point in a revised model figure and accompanying text, as discussed in the response to the next comment.

11. The model presented in Fig. 6 does not seem to summarize the obtained data, and does not help in the understanding of the new concept(s). The authors should either remove it or present it in a more comprehensive way.

We modified the original Fig 6 (now Fig 5) to better explain the model and to make the proposal more intuitive. We added panels A and B illustrating the interpretation of residence time estimates that led us to the final summary model (now panel C). We also added the following text to the Fig 5 legend:

A,B Interpretation of residence times. (A) A GTF can undergo multiple rounds of transient binding (indicated by x and shown in light blue) before it binds stably (dark blue), possibly assisted by the other factors (gray). (B) The final residence time estimates at a given site represent the average of transient and stable binding over the course of the experiment across all cells. Multiple transient binding events over time are shown in light blue; stable binding events block sites from exchange and are shown in dark blue. X-bar denotes the final residence time estimate derived from transient and stable binding events.

12. The authors' model works only if the CC method detects a real exchange between the "old" and the "new" factor. What if the induced protein signal increases, even when the residing protein signal does not decrease and remains constant? Thus, it would not be an exchange nor a competition. Can the authors show that the "old" factor ChIP signal decreases, when the newly synthesized factor arrives?

The reviewer makes an interesting point. We first addressed these questions by simulating turnover dynamics using the turnover model, which is a physically rigorous mass action model

of turnover dynamics, under realistic occupancy assumptions, namely, relatively low occupancy (e.g., occupancy $\sim 10\%$ with respect to accessible sites on DNA; this is different for TFIIE as detailed below). More specifically, we simulate TBP assuming the residence time is 5 min (i.e., $t_{1/2} = 5$ min) or the off-rate is 0.14 min^{-1} (i.e., $k_d = 0.14 \text{ min}^{-1}$), the concentration of HA-TBP and Myc-TBP are the same at steady state (i.e., $t \rightarrow \infty$) and the overall on-rate is 0.017 min^{-1} (i.e., $k_{a\text{CTBP}} = 0.017 \text{ min}^{-1}$). This will lead to the occupancy of HA-TBP and Myc-TBP to both be 0.1 at steady state (i.e., $t \rightarrow \infty$). As shown below, the occupancy of HA-TBP rises from 0 to 0.1 over time post induction, as expected. While we find that the occupancy of Myc-TBP drops, it only drops by 10%. This drop would be relatively difficult to detect using ChIP-based approaches. Consequently, we may observe relatively flat Myc-TBP profiles; however, our approach only requires that the HA-TBP/Myc-TBP ratio saturates at different times for different sites to enable distinct residence times to be estimated.

Thus, the extent to which one would see a change in the signal of the Myc isoform depends on the fractional occupancy of the GTF as well as the binding kinetics, and modest changes in Myc isoform signal, when observed, are well within expectation. Given that we also observe evidence for effectively higher occupancy sites (i.e., with respect to complexes that form infrequently on which TFIIE binds), we also simulated effective higher occupancy by keeping all the parameters above the same except the overall on-rate, $k_{a\text{CTFIIE}}$, which we set to 0.1 min^{-1} . This corresponds to a 30% drop in effective occupancy for endogenous Myc-TFIIE from $t = 0$ min to $t \rightarrow \infty$ consistent with the median drop in Myc-TFIIE normalized counts shown below.

To confirm that the models apply to the data measured by us, we analyzed the normalized Myc and HA ChIP signal within promoter sites (results presented in the box plots below). The data presented here align well with the models, where we observed drops in Myc signal consistent with low and effectively high occupancy sites. Intuitively, the more pronounced decrease in Myc signal over time for TFIIIE and TFIIIF compared to other GTFs is consistent with our model and the prediction that there would be competition for effectively high occupancy binding at complexes that form infrequently. However, a rigorous test of this idea is we feel beyond the scope of the present study.

We have provided these data to the reviewers but not included them in the revision because of our concern that the results could be a distraction. Importantly, this analysis now begins to touch on the overall on-rate which requires accurate relative measurements of the concentration of the HA- and Myc-tagged GTF proteins as well as a more complex turnover model analysis, which is a goal for future work. However, if the reviewer feels such information is important to include, we would be more than happy to add it to the paper.

Reviewer 2

Specific comments:

1. The Buratowski lab (Baek et al. 2021, Mol Cell) has used single-molecule TIRF microscopy to monitor the kinetics of GTF binding to a model gene in vitro and observed frequent dissociation of TFIIE over the time course of their experiments. They concluded that TFIIE association was unexpectedly dynamic. The authors have not cited this paper or provided an explanation for the apparent discrepancy.

We apologize for the oversight in not citing this important paper and we have now included it in the revision in the discussion (lines 307-317):

The broad outlines of the pathway suggested here are compatible with the notion of dynamic and even branching assembly pathways proposed on the basis of observations of single complexes formed using nuclear extracts (Baek et al, 2021). As the work presented here includes a genome-scale inventory of kinetic behavior and most promoters do not possess a regulatory region (Rossi et al, 2021), the distribution of residence times reflects the behavior of GTFs at such promoters. This probably explains why, for example, we observe promoters with a wide range of TFIIE residence times (including promoters where it is relatively long-lived) whereas Baek et al (2021) observed unexpectedly dynamically bound TFIIE in that system. Additionally, our data do not have sufficient resolution to distinguish GTF loading at regulatory regions (Baek et al, 2021) versus core promoters, and we are therefore unable to draw inferences about the impact of activators per se on the assembly process.

We have also added citations for Baek et al (2021) and Nguyen et al (2021) to the introduction in which we review published evidence that the canonical in vitro assembly pathway may not explain PIC assembly at promoters *in vivo* (lines 62-65):

Furthermore, some evidence suggests that the canonical in vitro assembly pathway may not apply to PICs at all promoters in vivo (Guglielmi et al, 2013; Luse, 2014; Sikorski & Buratowski, 2009; Baek et al, 2021; Nguyen et al, 2021).

2. One other surprising aspect of the Baek et al. paper was the finding of GTFs at UAS elements. Are the authors' ChIP data of sufficient resolution to measure residence times at UAS elements?

This is a very good question! Unfortunately, and as noted above, we do not have the resolution to address it. The ChIP-seq DNA fragment lengths and peak sizes are on the same order as the average intergenic region size. It would be very interesting to address this using ChIPexo combined with the competition ChIP approach, but we feel that this is beyond the scope of the current study.

3. Because the ChIP and DTA experiments involve galactose induction, there is at least one group of genes undergoing temporal activation (the GAL genes). Can the authors comment on

the residence times of the GTFs at these genes and whether they differ from those for constitutively expressed genes in their data set?

Thank you for the suggestion. We compared residence times and synthesis rates between GAL genes (for which we have measurements) and all other genes (new Figure S6). While we observed significantly higher synthesis rates of the GAL genes, we did not see any significant differences in GTF residence times. We added the following text to the manuscript (lines 206-209):

In contrast, we did not observe any significant differences between the residence times of galactose-induced genes and all other genes, even though the synthesis rates of the GAL genes were significantly higher than the genes that were not induced by galactose (Appendix Fig S6A-B).

4. The use of the term Hill equation and Hill coefficient is confusing, as the Hill coefficient is most often used to describe binding cooperativity.

We clarify this point by modifying the sentence which first discusses the Hill equation and coefficient in the Results section in as follows (lines 106-108):

The time-dependent accumulation of competitor isoforms displayed cooperative induction consistent with a Hill equation (Estrada et al, 2016) with induction half-times of ~43 min and Hill coefficients of ~4.5 on average (Fig 2B,C).

5. By what measure did the authors confirm that the tags on the GTFs did not affect activities of the factors? Data showing the tags are functionally inert are needed.

In the revision, we include new figures showing the growth of yeast strain harboring the Myc- and HA-tagged alleles (Appendix Fig S9). To demonstrate the functionality of the Myc-tagged alleles, we show growth of the diploid competition strains whose growth on glucose is provided solely by the Myc isoform. On galactose, one can observe growth following induction of the HA-tagged isoform. To assess the HA-isoform more directly, we scored yeast growth of the haploid segregants that contain the HA-tagged subunit as the sole source of the GTF in these strains. (Note that in these haploid cells for TFIIA, TFIIIE, and TFIIIF, they also carry the untagged GTF subunit under *GAL1* control). These results demonstrate the functionality of the tagged isoforms in vivo.

This text was added to the revised manuscript to accompany the figure (lines 392-394):

Functionality of the Myc- and HA-tagged alleles was confirmed by spot tests and streaking of the strains on appropriate media (Appendix Fig S9).

6. Figure 2E and S2B--- Are these data from one replicate or an average of more than one replicate?

Each data point shown in Figures 2E and Fig S1B (originally Fig S2B) represents one replicate; however, the requirement that the normalized data from eight time points fit the Hill function and turnover model with $R^2 > 0.7$ functions as a form of replication. We clarify this point by adding the following sentence on lines 439-441:

While we only gathered one replicate competition ChIP-seq sample per time point, the normalized data from the eight time points were required to fit both the Hill model and turnover model with $R^2 > 0.7$ as detailed below.

7. Figure 2F--- Define "density" in the legend.

We added the following definition to the legend for Fig. 2F (lines 1092-1093):

Density on the y-axis denotes the kernel density estimates used to approximate the frequency of a given residence time.

8. Figure 3C--- This is a key result. Measures of statistical significance are needed.

We performed two sets of statistical tests:

- 1) Kruskal-Wallis tests were used to identify any significant changes in residence time across all four synthesis quartiles for each GTF, and
- 2) Wilcoxon tests were used to identify any significant changes in residence time between the indicated pairs of synthesis quartiles. We report the results of the Kruskal-Wallis tests in the form of p-values and those of the Wilcoxon tests in the form of p-value symbols in the revised figure, and they are described in the revised figure legend as follows (lines 1109-1111):

*P-values represent results of Kruskal-Wallis tests for a given GTF. P-value symbols (Wilcoxon tests): n.s. $p \geq 0.1$, . $p < 0.1$, * $p < 0.05$, ** $p < 0.01$, *** $p < 0.001$.*

We further updated the Materials and Methods section to reflect the changes (lines 669-671):

Normality was tested with q-q plots. Statistical testing between pairs of groups was performed using unpaired two-sided Wilcoxon tests, and overall summary for each GTF with Kruskal-Wallis test.

9. Figure 4 is introduced as a pair-wise comparison of the GTF residence times; however little discussion is given to these comparisons. What is the value of these GTF-GTF comparisons? Panels in S5 might be more relevant to include in the main body.

We decided upon further discussion that the figure does not warrant inclusion in the main body of the paper as it only adds to the results that are shown and discussed in other figures which we feel are more important. In the revision, the figure was moved to the Expanded View as Fig EV4. We also expanded the text associated with the figure as follows (lines 169-173):

This allowed us to identify a cluster of highly transcribed genes (dark cluster) associated with the presence of long-lived TFII E and TFII F, as well as TFII B residence times that were in a similar range of several minutes. This was in contrast to TBP and TFII A, whose residence times did not show any significant pattern.

In addition, we moved the original Fig S5 to the Expanded View section, where it is now Fig EV3.

10. Figure S1--- Any thoughts on why TFII A and TFII E levels accrue more slowly upon GAL induction than the other factors? Also, panel E in this figure requires a label on the x-axis.

Please see the response to comment (3) by Reviewer 1 above.

We added the x-axis label to what is now Fig EV1.

11. Figure 3 panels E-G vs. Figure S4 A-C. The distinctions between these figures should be made clearer to the reader.

We expanded the explanation and discussion of these figures in the text and highlighted the difference in (now) Figure S2 legend (lines 156-166):

Using all of the GTF residence time data for Principal Component Analysis (PCA) revealed a correlation between GTF binding dynamics and RNA synthesis along the first principal component, PC1 (Fig 3E; Appendix Fig S2, where sites with <1 min residence times, which were randomly generated between 0-1 min, were excluded). This correlation can be appreciated quantitatively via the proportion of variance explained and visually by the distribution of color across the plot. To investigate the nature of this relationship in more detail, Pearson's correlation coefficients were computed between each GTF and PC1/PC2 (Fig 3F) and between transcription rates and PC1/PC2 (Fig 3G). The results show that the overall pattern was driven mainly by the positive correlations between TFII E/TFII F and RNA synthesis rate (Fig 3F,G). This conclusion was further supported by linear modeling of the GTF residence time contributions to transcription rates (Fig EV3A,B).

Figure S2A: PCA plot separating genes (points) based on GTF residence times. Color coding indicates the mean synthesis rate of the genes falling under a given area. Note that in comparison to Fig 3E, in this figure genes with residence time estimates <1 min were excluded from the analysis.

Reviewer 3

- I noted that the authors expressed the other subunits of IIA, IIF and IIE simultaneously although I question why they felt that necessary. Clearly some IP and blots could have determined whether the HA tagged subunit readily exchanged into the dimeric GTFs.

It would have been possible to measure exchange between newly made competitor subunits and existing complexes, but it was unclear to us if we could have measured exchange kinetics with sufficient accuracy and time resolution to be confident that exchange was not an issue. For example, there could continue to be subunit exchange during a conventional IP and so time resolution of the exchange process could be challenging to assess by a standard or typical experiment. Certainly, such questions could be addressed experimentally but we felt that they would potentially take a lot of time and effort to do well. For these reasons, we felt it was simpler to express both subunits and avoid questions of exchange kinetics or potential competition between limiting subunits or complexes.

- Additionally, I had trouble understanding how the final GAL-stimulated levels of HA-GTF matched the total level of the HA+Myc GTF in cells, e.g., normal TBP level versus induced level as measured with a TBP antibody.

The levels of the HA isoforms likely do not precisely match the native levels, but this does not matter for the method. We do not have polyclonal antibodies on hand for all the GTF subunits, but for example, we observed that the GAL-induced HA-TBP isoform is expressed at nearly three-fold the level of Myc-TBP (our unpublished results). As detailed in Zaidi et al (Zaidi et al. 2017a reference in manuscript) and more briefly in this manuscript (see lines 526 – 534), because we are fitting the turnover model to the ratio of the GAL-induced HA-TBP isoform over Myc-TBP levels, we only need to normalize the CC data so that the $t \rightarrow 0$ and $t \rightarrow \infty$ boundary conditions are met. These are satisfied as long as the CC ratio data is approximately 0 at the zero time point and saturates to the same ratio as the measured protein levels. Notably, even if the protein ratios were re-normalized, all that is required to satisfy the boundary conditions is that the CC ratio data is normalized to the same re-normalized ratio levels as the protein data. Finally, another place where this issue would be important, in principle, is if we were attempting to derive the on-rate, k_a . However, we are not. As shown in Zaidi et al. and Lickwar et al. (Lickwar et al. 2012 reference in manuscript), the ratio of the GAL-induced HA-TBP isoform over Myc-TBP levels is insensitive to the on-rate, k_a . We further clarify this point by adding the following text at the beginning of the mass action modeling subsection in the Materials and Methods section (lines 509-515):

We then fit a mass action kinetic turnover model to the estimated ratio of fractional occupancies at every promoter site where a peak was identified to derive the residence time for a GTF at that site. As previously reported (Lickwar et al, 2012; Zaidi et al, 2017a) and detailed below, the ratio, $\theta_B(t)/\theta_A(t)$, is insensitive to the overall on-rate which is the only place that the concentrations of HA- and Myc-tagged proteins enters the mass action model. Consequently, the estimation of residence time is insensitive to the relative levels of HA- and Myc-tagged proteins at steady state.

- There was a normalization process used in the displacement assays but the level of overexpression of one GTF versus the other might influence the dynamics and although I am somewhat satisfied with the authors treatment of this issue, I was wondering whether the

rightward shift in the curves could be affected or correlated with higher concentrations of the final TFIIF and TFIIE levels versus TFIIB and TBP, which are single subunits.

As mentioned in the response above, Zaidi et al. and Lickwar et al. (Lickwar et al. 2012 reference in manuscript) showed that the ratio of the GAL-induced HA-GTF isoform over Myc-GTF levels is insensitive to the overall on-rate, $k_a c_{GTF}$. Importantly, the concentration of the GTFs only enters mass action turnover dynamics and the turnover model together with the on-rate in the form $k_a c_{GTF}$. Consequently, the rightward shift in the curves is highly insensitive to the overall on-rate, $k_a c_{GTF}$, hence the concentration of the GTF, c_{GTF} , for both the GAL-induced HA-GTF and Myc-GTF isoforms. The reviewer is absolutely correct that these differences in concentrations would affect the overall on-rate and would have to be carefully accounted for if an estimate of the molecular on-rate, k_a , were attempted using CC data. As mentioned above, we added new text at lines 509-515 to clarify this point.

- Would have liked to see some specific browser tracks displaying the dynamics for all the factors at the same promoter analogous to what was shown in Supplemental 1 but much more thorough. This would have been particularly interesting on the Ribosomal protein genes, where Tony Weil earlier showed an effect of Rap1 on the TFIIA-TFIID complex and argued for interactions between Rap1, TFIIA and TAF4 (PMC3743499). This is particularly interesting because others have shown in the human PIC that TFIIA interacts with TAF4 within the high res 3D structure. I have a hunch the TAFs might be a better indicator of TFIID occupancy.

This is an important point. We have added genome browser tracks for all the measured GTFs showing the log2 transformed HA/Myc CHIP signal ratios at multiple genes from heatmap cluster 1. These are primarily ribosomal subunit genes, but we also included *CDC19* and *GAL80* for comparison. The browser tracks are presented in a new supplemental figure, Fig S3. While we included genes with a broad range of GTF residence times, and one can see exchange visually, the quantitative differences in signal contributing to the different residence times are not readily observable by eye. For this reason, we focus on the statistically significant differences in the main body of the paper without further discussion (lines 196-199)

The longer GTF residence times (as well as higher gene expression rates) at ribosomal protein genes in cluster 1 compared to the GTF residence times at other genes are statistically highly significant (Fig EV5A-C, cluster 1 gene CC signal tracks shown in Appendix Fig S3).

Dear Dr. Auble,

Thank you for submitting a revised version of your manuscript. Your study has now been seen by all original referees, who find that their previous concerns have been addressed and now recommend publication of the manuscript. There remain only a few mainly editorial points that have to be addressed before I can extend formal acceptance of the manuscript:

1. DATA AVAILABILITY SECTION: in, subtitle should be renamed to "Data Availability"
2. COI: title needs renaming to "DISCLOSURE AND COMPETING INTERESTS STATEMENT"
3. AC/CRedit: section needs to be removed
4. DATASET EV LEGENDS: Table EV1-EV3 should be renamed to Dataset EV1-EV3 with the corresponding callouts; the legends should be uploaded as a separate tab in each Excel file
5. APPENDIX 1 FILE WITH ToC: nomenclature should be Appendix Figure S1-S9 and Appendix Table
6. Synopsis:

Papers published in The EMBO Journal are accompanied online by a 'Synopsis' to enhance discoverability of the manuscript. It consists of A) a short (1-2 sentences) summary of the findings and their significance, B) 3-4 bullet points highlighting key results and C) a synopsis image that is 550x300-600 pixels large (width x height, jpeg or png format). You can either show a model or key data in the synopsis image. Please note that the image size is rather small, and that text needs to be readable at the final size. Please send us this information together with the revised manuscript.

7. Section "Background" should be renamed to "Introduction"

Our data editors have flagged the following issues in figure legends that need correcting:

1. Figure Legends (main + EV): "1. Please indicate the statistical test used for data analysis in the legend of figure EV 5b.
2. Please note that in figure 3c; there is a mismatch between the annotated p values in the figure legend and the annotated p values in the figure file that should be corrected."
3. Please note that information related to n is missing in the legends of figures 3c-d; EV 5a-c.

With best regards,

Cornelius Schneider

Cornelius Schneider, PhD
Editor
The EMBO Journal
c.schneider@embojournal.org

- a point-by-point response to the referees' comments, with a detailed description of the changes made (as a word file).
- a word file of the manuscript text.

- individual production quality figure files (one file per figure)
 - a complete author checklist, which you can download from our author guidelines (<https://www.embopress.org/page/journal/14602075/authorguide>).
 - Expanded View files (replacing Supplementary Information)
- Please see out instructions to authors
<https://www.embopress.org/page/journal/14602075/authorguide#expandedview>

We realize that it is difficult to revise to a specific deadline. In the interest of protecting the conceptual advance provided by the work, we recommend a revision within 3 months (12th May 2024). Please discuss the revision progress ahead of this time with the editor if you require more time to complete the revisions. Use the link below to submit your revision:

Referee #1:

The authors have satisfactorily answered most of the concerns raised.

Referee #2:

The authors have been very responsive to previous reviewer comments. In the revised manuscript, the authors have more clearly described their results and better placed the conclusions in the context of several important publications on the kinetics of PIC assembly. Including the current study, the dynamics of PIC assembly have now been viewed through genome-wide competition ChIP, single molecule experiments on defined promoters, and single molecule tracking assays within live cells (among other studies). Differences in the conclusions likely relate to the strengths as well as limitations of each experimental approach. The current study by Kupkova et al. adds a new view to this important problem by measuring PIC assembly in vivo on a genomic scale. This is an important contribution.

Referee #3:

Dr. Auble answered all of my questions from the initial review. All the reviewers felt the study would have been improved by including Mediator, TFIID and TFIIH but I personally believed that such a request was almost technically unfeasible because of the necessity of co-expressing all of the subunits, as argued by Auble. Auble added a nice browser track with Gal80, CDC19 and some ribosomal protein genes that covered my points about using such visualization devices to support the major conclusion and I felt the data did just that. Otherwise my comments were mainly queries about interpretations, which Auble handled nicely. As before, I feel this is a really good tour de force paper that should be published without further delay. The amounts of quality data are breathtaking and his lab is to be congratulated for their skill and dedication.

Thank you for submitting a revised version of your manuscript. Your study has now been seen by all original referees, who find that their previous concerns have been addressed and now recommend publication of the manuscript. There remain only a few mainly editorial points that have to be addressed before I can extend formal acceptance of the manuscript:

1. DATA AVAILABILITY SECTION: in, subtitle should be renamed to "Data Availability"

The section has been renamed as requested.

2. COI: title needs renaming to "DISCLOSURE AND COMPETING INTERESTS STATEMENT"

The title has been changed as requested.

3. AC/CRedit: section needs to be removed

We removed the Author Contribution section.

4. DATASET EV LEGENDS: Table EV1-EV3 should be renamed to Dataset EV1-EV3 with the corresponding callouts; the legends should be uploaded as a separate tab in each Excel file

We renamed Table EV1-EV3 to Dataset EV1-EV3 and changed the references in the manuscript text accordingly.

5. APPENDIX 1 FILE WITH ToC: nomenclature should be Appendix Figure S1-S9 and Appendix Table

We changed the ToC in Appendix 1 to Appendix Figure and Appendix Table.

6. Synopsis:

Papers published in The EMBO Journal are accompanied online by a 'Synopsis' to enhance discoverability of the manuscript. It consists of A) a short (1-2 sentences) summary of the findings and their significance, B) 3-4 bullet points highlighting key results and C) a synopsis image that is 550x300-600 pixels large (width x height, jpeg or png format). You can either show a model or key data in the synopsis image. Please note that the image size is rather small, and that text needs to be readable at the final size. Please send us this information together with the revised manuscript.

We added the following synopsis text and figure to the submitted files:

The RNA polymerase II transcription machinery is understood in structural detail, but much less is known about its assembly dynamics on promoter DNA *in vivo*. Using competition chromatin immunoprecipitation, we measured the genome-scale chromatin residence times of five key general transcription factors (TBP, TFIIA, TFIIB, TFIIE, and TFIIIF) in budding yeast.

- Many interactions were short-lived (<1 min), however, interaction residence times in the several minutes range were also found for each factor.

- Genes with shared biological functions had promoters with shared chromatin binding kinetic behavior.

- TFIIE, which binds late in the assembly process, had residence times that correlated with the rates of RNA synthesis.

- These data provide a rich resource for exploring mechanistic relationships between transcription complex assembly dynamics and RNA production.

7. Section "Background" should be renamed to "Introduction"

The section has been renamed as requested.

Our data editors have flagged the following issues in figure legends that need correcting:

1. Figure Legends (main + EV): "1. Please indicate the statistical test used for data analysis in the legend of figure EV 5b.

We added the following text to the legend for Figure EV 5B:
 "Wilcoxon p-value is indicated."

2. Please note that in figure 3c; there is a mismatch between the annotated p values in the figure legend and the annotated p values in the figure file that should be corrected."

We corrected the p-value annotations in the legend to Figure 3C. The text now states:
 "P-value symbols (Wilcoxon test): n.s. $p \geq 0.05$, * $p \leq 0.05$, ** $p \leq 0.01$, *** $p < 0.001$, **** $p \leq 0.0001$."

3. Please note that information related to n is missing in the legends of figures 3c-d; EV 5a-c.

We added the missing information about n to the figure legends:

Figure 3C: "Number of observations (n): TBP- Q1: 216, Q2: 324, Q3: 427, Q4: 605; TFIIA- Q1: 239 , Q2: 320, Q3: 397, Q4: 426; TFIIB Q1: 302, Q2: 486, Q3: 581, Q4: 606; TFIIF- Q1: 253, Q2: 358, Q3: 432, Q4: 517; TFIIE- Q1: 416, Q2: 584, Q3: 610, Q4: 553."
 Figure 3D: "Number of observations (n): TBP = 1572, TFIIA = 1382, TFIIB = 1975, TFIIF = 1560, TFIIE = 2163."

Figure EV3A: "Number of observations (n): not ribosomal/ribosomal TBP- 2810/87; TFIIA- 2418/76; TFIIB- 3420/93; TFIIF- 2776/88; TFIIIE- 3723/86."

Figure EV3B: "Number of observations (n): not ribosomal- 3169, ribosomal- 57."

Figure EV3C: "Number of observations (n): not ribosomal/ribosomal TBP- Q1: 209/11, Q2: 320/10, Q3: 414/11, Q4: 583/14; TFIIA- Q1: 231/13 , Q2: 316/9, Q3: 382/13, Q4: 409/9; TFIIB Q1: 293/14, Q2: 475/12, Q3: 569/11, Q4: 589/12; TFIIF- Q1: 245/11, Q2: 355/12, Q3: 419/12, Q4: 494/12; TFIIIE- Q1: 403/13, Q2: 577/12, Q3: 598/10, Q4:540/10."

Dear Dr. Auble,

I am pleased to inform you that your manuscript has been accepted for publication in the EMBO Journal.

Yours sincerely,

Cornelius Schneider, PhD
Editor
The EMBO Journal
c.schneider@embojournal.org
